# Downregulation of the tyrosine degradation pathway extends *Drosophila* lifespan

Andrey A Parkhitko[1,2]*, Divya Ramesh[3,4], Lin Wang[5,6], Dmitry Leshchiner[1], Elizabeth Filine[1], Richard Binari[1,7], Abby L Olsen[8], John M Asara[9], Valentin Cracan[10,11], Joshua D Rabinowitz[5,6], Axel Brockmann[3], Norbert Perrimon[1,7]*

[1]Department of Genetics, Blavatnik Institute, Harvard Medical School, Boston, United States; [2]Aging Institute of UPMC and the University of Pittsburgh, Pittsburgh, United States; [3]National Centre for Biological Sciences, Tata Institute of Fundamental Research, Bangalore, India; [4]Department of Biology, University of Konstanz, Konstanz, Germany; [5]Department of Chemistry, Princeton University, Princeton, United States; [6]Lewis-Sigler Institute for Integrative Genomics, Princeton University, Princeton, United States; [7]Howard Hughes Medical Institute, Boston, United States; [8]Department of Neurology, Brigham and Women's Hospital, Massachusetts General Hospital, Harvard Medical School, Boston, United States; [9]Division of Signal Transduction, Beth Israel Deaconess Medical Center, and Department of Medicine, Harvard Medical School, Boston, United States; [10]Scintillon Institute, San Diego, United States; [11]Department of Chemistry, The Scripps Research Institute, La Jolla, United States

*For correspondence:
aparkhitko@pitt.edu (AAP);
perrimon@receptor.med.harvard.edu (NP)

Competing interests: The authors declare that no competing interests exist.

**Abstract** Aging is characterized by extensive metabolic reprogramming. To identify metabolic pathways associated with aging, we analyzed age-dependent changes in the metabolomes of long-lived *Drosophila melanogaster*. Among the metabolites that changed, levels of tyrosine were increased with age in long-lived flies. We demonstrate that the levels of enzymes in the tyrosine degradation pathway increase with age in wild-type flies. Whole-body and neuronal-specific downregulation of enzymes in the tyrosine degradation pathway significantly extends *Drosophila* lifespan, causes alterations of metabolites associated with increased lifespan, and upregulates the levels of tyrosine-derived neuromediators. Moreover, feeding wild-type flies with tyrosine increased their lifespan. Mechanistically, we show that suppression of ETC complex I drives the upregulation of enzymes in the tyrosine degradation pathway, an effect that can be rescued by tigecycline, an FDA-approved drug that specifically suppresses mitochondrial translation. In addition, tyrosine supplementation partially rescued lifespan of flies with ETC complex I suppression. Altogether, our study highlights the tyrosine degradation pathway as a regulator of longevity.

## Introduction

Aging is the primary risk factor for many major human pathologies, including cancer, diabetes, cardiovascular disorders, and neurodegenerative diseases (*López-Otín et al., 2013*). Untargeted and targeted metabolomics analysis in worms (*Fuchs et al., 2010*), flies (*Hoffman et al., 2014*; *Avanesov et al., 2014*; *Parkhitko et al., 2016*), mice (*Tomás-Loba et al., 2013*), and humans (*Yu et al., 2012*) have documented changes in the metabolome during the aging process. Manipulations of metabolic pathways that change with age might suppress aging and extend lifespan

(*Parkhitko et al., 2020*). For example, perturbation of mitochondrial function (*Copeland et al., 2009*; *Owusu-Ansah et al., 2013*; *Fridell et al., 2005*), activation of the pentose phosphate pathway (*Legan et al., 2008*), suppression of purine nucleotide metabolism (*Stenesen et al., 2013*), suppression of fatty acid oxidation (*Mourikis et al., 2006*), and inhibition of glycogen metabolism (*Sinadinos et al., 2014*; *Post et al., 2018*) have been shown to extend lifespan. Moreover, interventions that are known to extend lifespan, like dietary restriction, genetic selection or manipulations of specific pathways might reverse age-dependent metabolic reprogramming (*Laye et al., 2015*). In addition, key master regulators of metabolism such as DILPs/Insulin signaling (*Post et al., 2019*; *Bai et al., 2012*; *Broughton et al., 2005*; *Grönke et al., 2010*; *Clancy et al., 2001*), Tor (*Kapahi et al., 2004*), AMPK (*Ulgherait et al., 2014*; *Burkewitz et al., 2014*), JNK (*Wang et al., 2003*), Spargel (*Rera et al., 2011*), Nrf2 (*Sykiotis and Bohmann, 2008*), Activin/TGFβ (*Bai et al., 2013*), and Sirt4 (*Wood et al., 2018*) are known to extend lifespan, although whether they extend lifespan via their effect on metabolism or other processes is unknown. While numerous studies have shown that metabolism changes with age and the role of several metabolic pathways in aging has been characterized, we do not completely understand what drives these metabolic alterations and how they affect other biological processes.

Amino acids serve as building blocks for proteins and fuel different metabolic pathways. In addition to a well-known role of amino acids in lifespan extension by dietary restriction, manipulating metabolism of specific amino acids can extend lifespan in flies and other organisms. For example, methionine restriction (*Lee et al., 2014*) or activation of methionine flux (*Parkhitko et al., 2016*; *Parkhitko et al., 2019*) prolongs health and lifespan in flies and other species. Increased homocysteine processing via overexpression of cystathionine β-synthase (*dCBS*) extends *Drosophila* lifespan and is required for the beneficial effects of dietary restriction (*Kabil et al., 2011*). Genetic and pharmacological impairment of the tryptophan/kynurenine pathway promotes *Drosophila* lifespan (*Oxenkrug et al., 2012*) and tryptophan restriction extends rat lifespan (*Ooka et al., 1988*). Impairing threonine catabolism via glycine-C-acetyltransferase suppression promotes *Caenorhabditis elegans* lifespan (*Ravichandran et al., 2018*). Glycine supplementation can extend lifespan in both *C. elegans* (*Liu et al., 2019*) and mice (*Miller et al., 2019*). Despite the accumulating evidence, the role of non-proteogenic metabolism of specific amino acids in the regulation of aging and lifespan and their mechanisms are still poorly characterized.

One approach to identify new traits responsible for aging is to compare how these traits change with age in control and long-lived animals of the same species (*Milman and Barzilai, 2016*). For example, centenarians have a distinctive epigenetic profile compared to an age-matched control population (*Horvath et al., 2015*). Similarly, we previously showed that flies with increased longevity have dramatic differences in many metabolites associated with methionine metabolism even at 1 week of age when 100% of both control- and long-lived flies are still alive (*Parkhitko et al., 2016*). To identify novel metabolic pathways that correlate with lifespan and that can be responsible for aging, we compared the metabolome of 1-week- and 4-week-old wild-type and long-lived flies to identify changes in metabolites that correlate with lifespan and identified tyrosine as an age-dependent metabolite. We demonstrate that *Drosophila* has a single tyrosine aminotransferase (TAT). Whole-body or neuronal-specific downregulation of TAT as well as other downstream enzymes in the tyrosine degradation pathway significantly extend *Drosophila* lifespan, cause alterations of multiple metabolites associated with increased lifespan, and lead to an increase in tyrosine and tyrosine-derived neuromediators (dopamine, octopamine, and tyramine). We further demonstrate that mitochondrial dysfunction may serve as an age-dependent stimulus that redirects tyrosine from neuromediator production into mitochondrial metabolism. In conclusion, our studies highlight the important role of the tyrosine degradation pathway and position TAT as a link between neuromediator production, dysfunctional mitochondria, and aging.

## Results

### Age-dependent changes in tyrosine levels

We previously demonstrated that many metabolites associated with methionine metabolism, a metabolic pathway playing a key role in regulation of aging, are dramatically different between 1-week-old wild-type and long-lived flies (*Parkhitko et al., 2016*). These long-lived flies have been selected

for delayed reproductive senescence over 170 generations and maintained on a generation interval of 70 days, while control lines were maintained on a 2-week generation interval (*Carnes et al., 2015*). To further extend our metabolic analysis and reveal new metabolites involved in the regulation of aging and lifespan, we compared differences in metabolomes in 1-week and 4-week-old wild-type (B3) and long-lived (O1 and O3) flies, searching for metabolites that are either different between control vs. long-lived flies of the same age and/or metabolites that change differently with age between control vs. long-lived flies. Although longevity is not the only trait that is different between these lines and these metabolites can be linked to other traits such as reproduction; we used this list of candidate metabolic pathways for the further analysis in wild-type flies. Including metabolites in methionine metabolism, we identified 49 metabolites whose changes were significantly different with age between control and both lines of long-lived flies (*Figure 1A*). While many of these metabolites belong to the metabolic pathways that have been previously described as being important players in the regulation of aging and longevity (tryptophan metabolism, NADPH, nucleotide metabolism etc.), many of them have not been studied before, including tyrosine.

Interestingly, levels of tyrosine significantly increased with age in both lines of long-lived flies (*Figure 1B*). In addition to its proteogenic function, tyrosine can be used either as a precursor for the synthesis of neuromediators, such as dopamine, octopamine, and tyramine, or can be degraded via the tyrosine degradation pathway producing acetoacetate and fumarate (*Figure 1E*). The first and rate-limiting enzyme in the tyrosine degradation pathway is tyrosine aminotransferase (TAT). Based on the DIOPT ortholog prediction tool (*Hu et al., 2011*), *Drosophila* has a single *TAT* ortholog, *CG1461*. We tested whether levels of *CG1461/TAT* were different between control and long-lived flies. Strikingly, mRNA levels of *CG1461/TAT* were significantly decreased in both long-lived O1 and O3 female and male flies compared to B3 control flies (*Figure 1C*), suggesting that TAT levels may explain age-dependent differences in the level of tyrosine.

Next, we examined *CG1461/TAT* expression levels in young (1 w) vs. old (5 w) wild-type *Oregon R* (*OreR*) flies and detected a significant mRNA increase in older flies (*Figure 1D*). Consistent with this observation, using a GFP-tagged transgenic line from the fly-TransgeneOme (fTRG) library (*Sarov et al., 2016*), we observed an increase in *TAT* with age (*Figure 1—figure supplement 1*). In addition, mRNA levels of other enzymes in the tyrosine degradation pathway, *Hgo*, *CG11796*, *GstZ2*, and *Hn*, were significantly increased with age in wild-type flies (*Figure 1D,E*). Altogether, these findings suggest that tyrosine catabolism increases during aging.

## Supplementing tyrosine extends lifespan and CG1461 functions as TAT to cope with high tyrosine levels

TAT catalyzes the conversion of tyrosine to 4-hydroxyphenylpyruvate and is the first and rate-limiting enzyme in the tyrosine catabolic pathway (*Figure 1E*). We first tested whether the *Drosophila* ortholog of *TAT*, *CG1461*, is involved in tyrosine degradation. We confirmed the knockdown efficiency of three independent *CG1461* RNAi lines by qRT-PCR (*CG1461* RNAi-1, ~50% [weak line]; *CG1461* RNAi-2 and 3, ~80% [strong lines]) (*Figure 2A*) and generated *CG1461*-deficient flies using CRISPR/Cas9 (*Figure 2B*). To test the functional significance of *CG1461* for tyrosine degradation, we ubiquitously downregulated *CG1461* in adult flies using the *tubulin-Gal4, tubulin-Gal80ts* temperature-inducible system. Gal80ts is active at 18°C and represses Gal4, whereas at 29°C, Gal80ts is inactivated, allowing Gal4-dependent expression of *CG1461* RNAi. Flies were grown at 18°C, switched to 29°C after eclosion to induce expression of *CG1461* RNAi and after 2 weeks switched on food supplemented with 5 g/L of tyrosine, which represents approximately a 5- to 10-fold increase of tyrosine compared to regular food (*Piper, 2017*). Extra tyrosine caused acute toxicity and death when *CG1461* expression was downregulated and the rate of death correlated with the strength of the RNAi lines, suggesting that CG1461 is critical in the degradation of excess tyrosine (*Figure 2C*). Similarly, when we switched *CG1461* wild-type, heterozygous, or mutant flies on food supplemented with 5 g/L of tyrosine it caused acute toxicity and death of *CG1461* mutant but no acute toxicity was observed in wild-type or heterozygous flies (*Figure 2D*). We further tested a range of different concentrations of tyrosine (1X, 2.5X, 5X, and 10X) and observed a gradual and significant decrease in the lifespan of mutant flies (−53%, −54%, −39%, −20% for 10X, 5X, 2.5X, and 1X concentrations of tyrosine in male flies and −69%, −67%, −48%, −37% for 10X, 5X, 2.5X, and 1X concentrations of tyrosine in female flies, respectively) (*Figure 2—figure supplement 1A,B*); while the lifespan was either non-affected in male wild-type flies (*Figure 2—figure supplement 1A*), or decreased at the

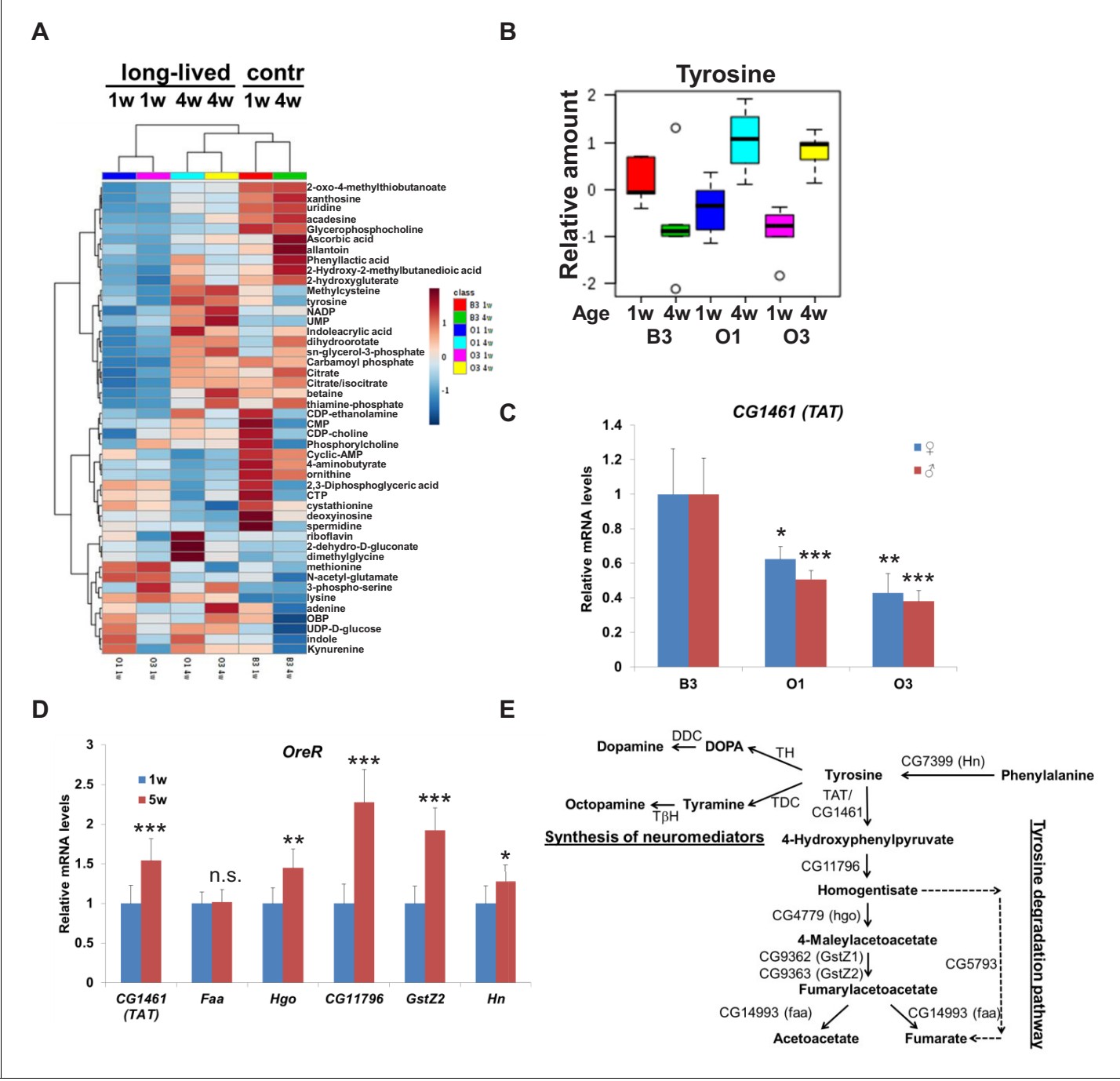

**Figure 1.** Tyrosine is a new lifespan-dependent metabolite. (**A**) Heat map showing the metabolites that significantly changed in 1-week and 4-week-old wild-type (**B3**) and long-lived (**O1** and **O3**) flies. Each row represents a mean of five biological replicates. (**B**) Box plots of relative levels of tyrosine in 1-week and 4-week-old wild-type (**B3**) and long-lived (**O1** and **O3**) flies extracted from the heat map (**A**). (**C**) Relative mRNA levels of *CG1461* in 1-week-old control (**B3**) and long-lived (**O1, O3**) flies. Means ± SD. (**D**) Relative mRNA levels of *CG1461*, *Faa*, *Hgo*, *CG11796*, *GstZ2, and Hn* from 1-week and 5-week-old wild-type (*OreR*) flies. Means ± SD. (**E**) Tyrosine metabolism pathway. *p<0.05, **p<0.01, ***p<0.001.

The online version of this article includes the following figure supplement(s) for figure 1:

**Figure supplement 1.** Age-dependent increase of GFP-tagged CG1461.

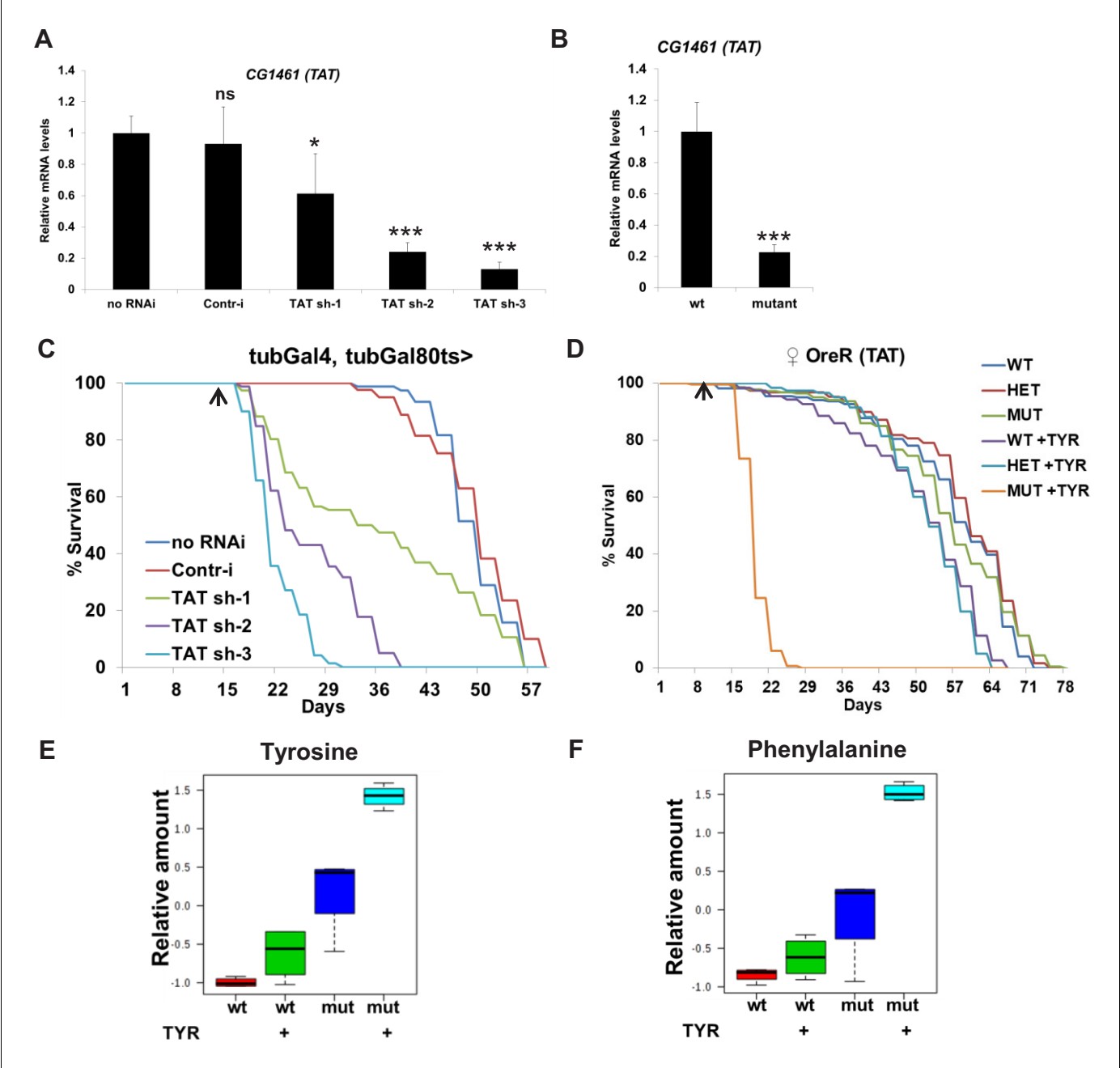

**Figure 2.** CG1461 functions as Tyrosine Aminotransferase/TAT and is necessary to degrade tyrosine. (A) Relative mRNA levels of *CG1461* in *tubulin-Gal80ts, tubulin-Gal4* flies expressing either no RNAi, control RNAi or three different *CG1461* RNAi for 10 days. Means ± SD. *p<0.05, ***p<0.001 (B) Relative mRNA levels of *CG1461* in wild-type and *CG1461*-deficient flies (both backcrossed to wild-type *OreR* flies). Means ± SD. ***p<0.001 (C) Feeding adult flies with 5 g/L of tyrosine significantly suppresses lifespan of flies with ubiquitous adult-onset downregulation of *CG1461*. Arrow indicates the beginning of tyrosine feeding. p<0.001. (D) Feeding adult flies with 5 g/L of tyrosine significantly suppresses lifespan of *CG1461* mutant but not wild-type or heterozygous flies. Arrow indicates the beginning of tyrosine feeding. p<0.001. Box plots of relative levels of tyrosine (E) and phenylalanine (F) in wild-type and *CG1461*-deficient flies fed either control or high level of tyrosine (5 g/L) diet.

The online version of this article includes the following figure supplement(s) for figure 2:

**Figure supplement 1.** Tyrosine supplementation increases lifespan of wild-type *OreR* flies and increases expression of GFP-tagged CG1461.

highest (5X and 10X) concentrations by a much lower extent in female wild-type flies (−38% and −24% for 10X and 5X concentrations of tyrosine) (*Figure 2—figure supplement 1B*). In addition, accordingly with the beneficial role of tyrosine, feeding flies with low concentrations of tyrosine significantly increased their lifespan by 9% in male flies (1X concentration of tyrosine, p<0.0001, log-rank test) (*Figure 2—figure supplement 1A*) and by 11.5% in female flies (1X concentration of tyrosine, p=0.002, log-rank test) (*Figure 2—figure supplement 1B*). Tyrosine is a conditionally essential amino acid because it can be synthetized from phenylalanine (but not vice versa). To examine whether *CG1461* regulates levels of tyrosine and phenylalanine, we performed metabolomic profiling of *CG1461* wild-type and mutant flies fed with either control or high-tyrosine diets. Feeding control flies with tyrosine increased the level of tyrosine and phenylalanine because phenylalanine is degraded via its conversion into tyrosine (*Figure 2E,F*); however, *CG1461* mutant flies on control diet had significantly higher levels of tyrosine, suggesting that a significant portion of tyrosine from regular food undergoes degradation via the tyrosine degradation pathway. Moreover, feeding *CG1461* mutant flies with tyrosine further increased levels of tyrosine and phenylalanine (*Figure 2E, F*) and caused their death (*Figure 2D*). We further tested whether the level of a GFP-tagged CG1461 changed in response to high tyrosine feeding, as we previously observed increased levels of *CG1461* with age. Feeding flies with 5 g/L of tyrosine increased the level of GFP-tagged CG1461 (*Figure 2—figure supplement 1C*). Interestingly, male flies had higher levels of GFP-tagged CG1461 both on control and tyrosine supplemented food (*Figure 2—figure supplement 1C*) that may explain the differences in response to the high concentration of tyrosine in male and female flies (*Figure 2—figure supplement 1A,B*). Moreover, mRNA levels of other enzymes from the tyrosine degradation pathway, *CG11796, Hgo, Faa, Hn,* were increased in male flies, whereas the level of *RP49* did not change (*Figure 2—figure supplement 1D*). Altogether, these data suggest that CG1461 is a functional ortholog of mammalian TAT that is required for the degradation of excess tyrosine to maintain stable levels of tyrosine on regular diet, and it is strongly induced when excess tyrosine is present. These data also point to the beneficial role of tyrosine in the regulation of lifespan in wild-type flies.

## The tyrosine degradation pathway regulates lifespan

Since supplementing tyrosine to wild-type flies increased their lifespan, levels of TAT and other enzymes in the tyrosine degradation pathway increased with age, and the levels of tyrosine were higher in the long-lived flies; we evaluated the effects of TAT and other enzymes in the tyrosine degradation pathway on lifespan using RNAi. To avoid developmental effects and differences in genetic backgrounds, we used the Actin Gene-Switch (Actin-GS) inducible Gal4/UAS expression system (*Roman et al., 2001*; *Osterwalder et al., 2001*), whereby UAS-RNAi expression is driven by Gal4 when flies are fed mifepristone (RU486). Expression of different control RNAi lines did not affect lifespan (*Parkhitko et al., 2016*), while two independent RNAi lines against *TAT* significantly extended lifespan (*TAT* RNAi-1 (weak), 9% increase in Mean Lifespan, p<0.0001, log-rank test; *TAT* RNAi-3 (strong), 17% increase in Mean Lifespan, p<0.0001, log-rank test) (*Figure 3A,B*), and better RNAi efficiency was consistent with more robust lifespan extension. While downregulation of TAT can result in lifespan extension due to a general inhibition of the tyrosine degradation pathway, it can also be due to a potential (although unknown) alternative function of TAT. To confirm that modulation of tyrosine degradation was responsible for lifespan extension, we downregulated two additional enzymes: *CG11796* and *Hgo*. CG11796 is a 4-hydroxyphenylpyruvate dioxygenase that catalyzes the conversion of 4-hydroxyphenylpyruvate to homogentisate, the second step in the tyrosine degradation pathway (*Figure 1E*). Hgo is a homogentisate 1,2-dioxygenase that catalyzes the conversion of homogentisate to 4-maleylacetoacetate (*Figure 1E*). Downregulation of both *CG11796* (*Figure 3C*) and *Hgo* (*Figure 3D*) moderately but significantly increased lifespan (*CG11796* RNAi, 10% increase in Mean Lifespan, p<0.0001, log-rank test; *Hgo* RNAi, 12% increase in Mean Lifespan, p<0.0001, log-rank test).

To further dissect the role of TAT in lifespan extension, we next tested whether specific tissues were responsible for lifespan extension. Expression of a strong *TAT* RNAi in the entire fat-body of adult flies starting at 1 week of age using Geneswitch driver strain WB-FB-GS, which contains both a head fat-body driver (S1-32) and a body-fat-body driver (S1-106) (*Giannakou et al., 2007*; *Shen et al., 2009*), did not affect lifespan (*Figure 3E*). Similarly, expression of strong *TAT* RNAi using the TIGS-2 Geneswitch driver (TIGS-2), which is associated with digestive tract-specific

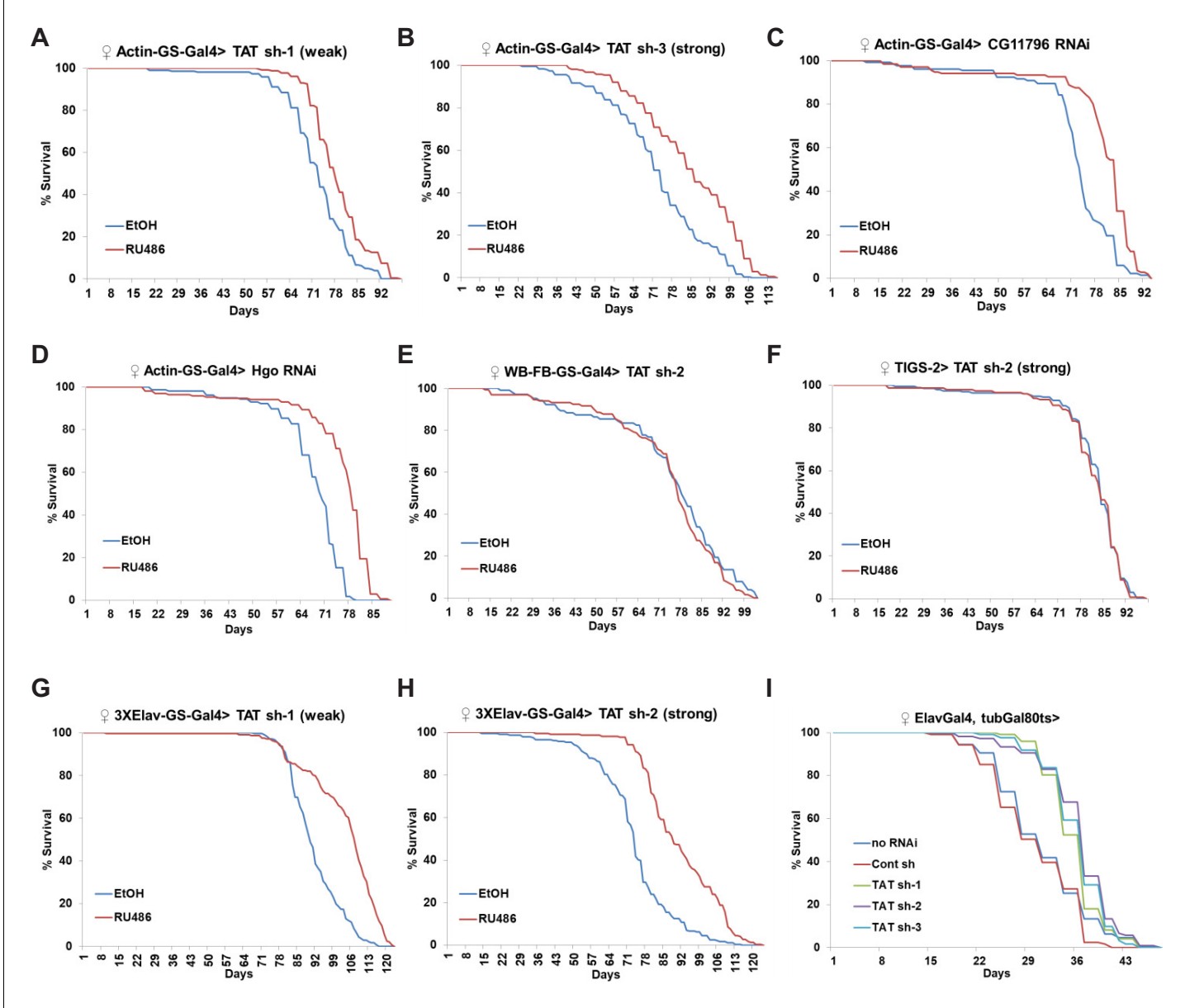

**Figure 3.** Whole-body and neuronal-specific downregulation of CG1461/Tyrosine Aminotransferase extends lifespan. (**A**) Ubiquitous adult-onset expression of *CG1461* RNAi-1 increases lifespan in females. p<0.0001. (**B**) Ubiquitous adult-onset expression of *CG1461* RNAi-3 increases lifespan in females. p<0.0001. (**C**) Ubiquitous adult-onset expression of *CG11796* RNAi increases lifespan in females. p<0.0001. (**D**) Ubiquitous adult-onset expression of *Hgo* RNAi increases lifespan in females. p<0.0001. (**E**) Fat body-specific adult-onset expression of *CG1461* RNAi does not affect lifespan in females. (**F**) Intestine-specific adult-onset expression of *CG1461* RNAi does not affect lifespan in females. (**G**) Neuronal-specific adult-onset expression of *CG1461* RNAi-1 increases lifespan in females. p<0.0001. (**H**) Neuronal-specific adult-onset expression of *CG1461* RNAi-2 increases lifespan in females. p<0.0001. (**I**) Neuronal-specific adult-onset expression of *CG1461* RNAi-1, -2, and -3 increases lifespan in females. p<0.0001. The online version of this article includes the following figure supplement(s) for figure 3:

**Figure supplement 1.** Neuronal-specific downregulation of Tyrosine Aminotransferase/TAT leads to metabolic reprogramming similar to the whole-body downregulation of TAT.

expression (*Rera et al., 2011*; *Poirier et al., 2008*), did not affect lifespan (*Figure 3F*). However, the 3XElav Geneswitch driver (3XElav-GS), which drives nervous-system-specific expression (*Osterwalder et al., 2001*; *Shen et al., 2009*), led to a significant extension of lifespan when two different *TAT* RNAi were expressed starting at 1 week (*TAT* RNAi-1 (weak), 12.6% increase in Mean Lifespan, p<0.0001, log-rank test; *TAT* RNAi-3 (strong), 24.5% increase in Mean Lifespan, p<0.0001,

log-rank test) (*Figure 3G,H*). Similar results were obtained when *TAT* RNAi lines were expressed in the nervous system of adult flies using ElavGal4 and the temperature-sensitive tubulin-Gal80ts repressor. Flies were allowed to develop at 18°C and then switched to 29°C after eclosion to induce RNAi expression. Adult onset neuronal-specific *TAT* RNAi expression resulted in strong lifespan extension compared to no RNAi or control RNAi expression (*TAT* RNAi-1, 15% increase in Mean Lifespan, p<0.0001, log-rank test; *TAT* RNAi-2, 18% increase in Mean Lifespan, p<0.0001; *TAT* RNAi-3, 17% increase in Mean Lifespan, p<0.0001, log-rank test) (*Figure 3I*). Interestingly, nervous-system-specific *TAT* downregulation resulted in stronger lifespan extension than with a ubiquitous driver. While differences in genetic background or/and the strength of Gal4 induction by mifepristone could explain these effects, one possibility is that *TAT* downregulation can be both beneficial and detrimental depending on tissue and cell type. Altogether, our results suggest that ubiquitous and tissue-specific suppression of the tyrosine degradation pathway is sufficient to extend lifespan.

## Ubiquitous downregulation of *TAT* causes metabolic reprogramming by affecting mitochondrial/antioxidant pro-longevity metabolic factors

To understand the mechanisms of lifespan extension by *TAT* downregulation, we performed metabolomic profiling of flies that expressed either control RNAi or two different strong *TAT* RNAi under the control of the ubiquitous temperature-sensitive (*tubulin-Gal4, tubulin-Gal80ts*) driver for 10 days. Principal component analysis (PCA) of the measured metabolites clearly distinguished flies with control and *TAT* RNAi but clustered the two independent *TAT* RNAi lines together (*Figure 4A*). We identified 24 metabolites that were significantly and commonly changed in flies expressing two different strong *TAT* RNAi compared to control flies (*Figure 4B*). As expected, one of the significantly altered metabolites was tyrosine (*Figure 4B,C*). Some of the other significantly changed metabolites have been previously connected to lifespan regulation. Normal aging and premature aging in mtDNA mutator mice exhibit increased brain lactate (*Ross et al., 2010*) and cerebrospinal fluid lactate is elevated in aging humans (*Yesavage et al., 1982*). Dietary supplementation with D-glucosamine-6-phosphate (GlcN-6-phosphate) extends lifespan of nematodes and aging mice acting as an inhibitor of glycolysis and promoting mitochondrial function (*Weimer et al., 2014*). Downregulation of *TAT* led to a significant increase of GlcN-6-phosphate (*Figure 4D*) and decrease in products of glycolysis - lactate (*Figure 4E*) and NADH (*Figure 4F*). Another lifespan-related metabolite that was significantly upregulated after downregulation of *TAT* was nicotinamide (*Figure 4J*). Nicotinamide supplementation improves healthspan in mice, reduces oxidative stress and inflammation (*Mitchell et al., 2018*). In flies, overexpression of *Nicotinamide mononucleotide adenylyltransferase (Nmnat)* and *Nicotinamidase (Naam)*, which encode enzymes involved in conversion of nicotinamide to NAD promotes longevity, improves mitochondrial function and protects against oxidative stress (*Balan et al., 2008*; *Liu et al., 2018*). Moreover, downregulation of *TAT* led to a significant increase of methylcysteine (*Figure 4G*) and decrease of the oxidized form of methionine, methionine sulfoxide (target of the MSRA antioxidant system). Dietary supplementation with *S*-methyl-*L*-cysteine has been shown to enhance the MSRA antioxidant system in *Drosophila* and to delay the progression of the movement defect in flies overexpressing α-synuclein in the nervous system (*Wassef et al., 2007*). Strikingly, five of the significantly altered metabolites, NADH, thiamine pyrophosphate, NADP, glutathione and lactate, belong to the pyruvate metabolism (*Figure 4B,E,F,I,K*) and point to metabolic pathways associated with mitochondrial function. Pyruvate is ultimately destined for transport into mitochondria as a major fuel to drive ATP production by oxidative phosphorylation and feed into multiple biosynthetic pathways intersecting the TCA cycle. The major sources of pyruvate in the cytoplasm are phosphoenolpyruvate, alanine and lactate (*Gray et al., 2014*).

We further tested whether neuronal-specific downregulation of *TAT* would cause metabolic alterations in whole flies similar to the whole-body downregulation of *TAT*. We performed metabolomics of flies that expressed either control RNAi or strong *TAT* RNAi (shRNA-2) under the control of the neuronal-specific temperature sensitive (*elav-Gal4, tubulin-Gal80ts*) driver for 10 days. Neuronal-specific downregulation of TAT caused the upregulation of tyrosine (*Figure 3—figure supplement 1A*) and metabolic reprogramming similar to the whole-body downregulation of TAT. We detected several metabolites (glucosamine, methylcysteine, and NADH) that were similarly altered as a result of whole-body-driven and neuronal-specific downregulation of *TAT* (*Figure 3—figure supplement 1B–D*). Altogether, our results suggest that downregulation of *TAT* causes global metabolic

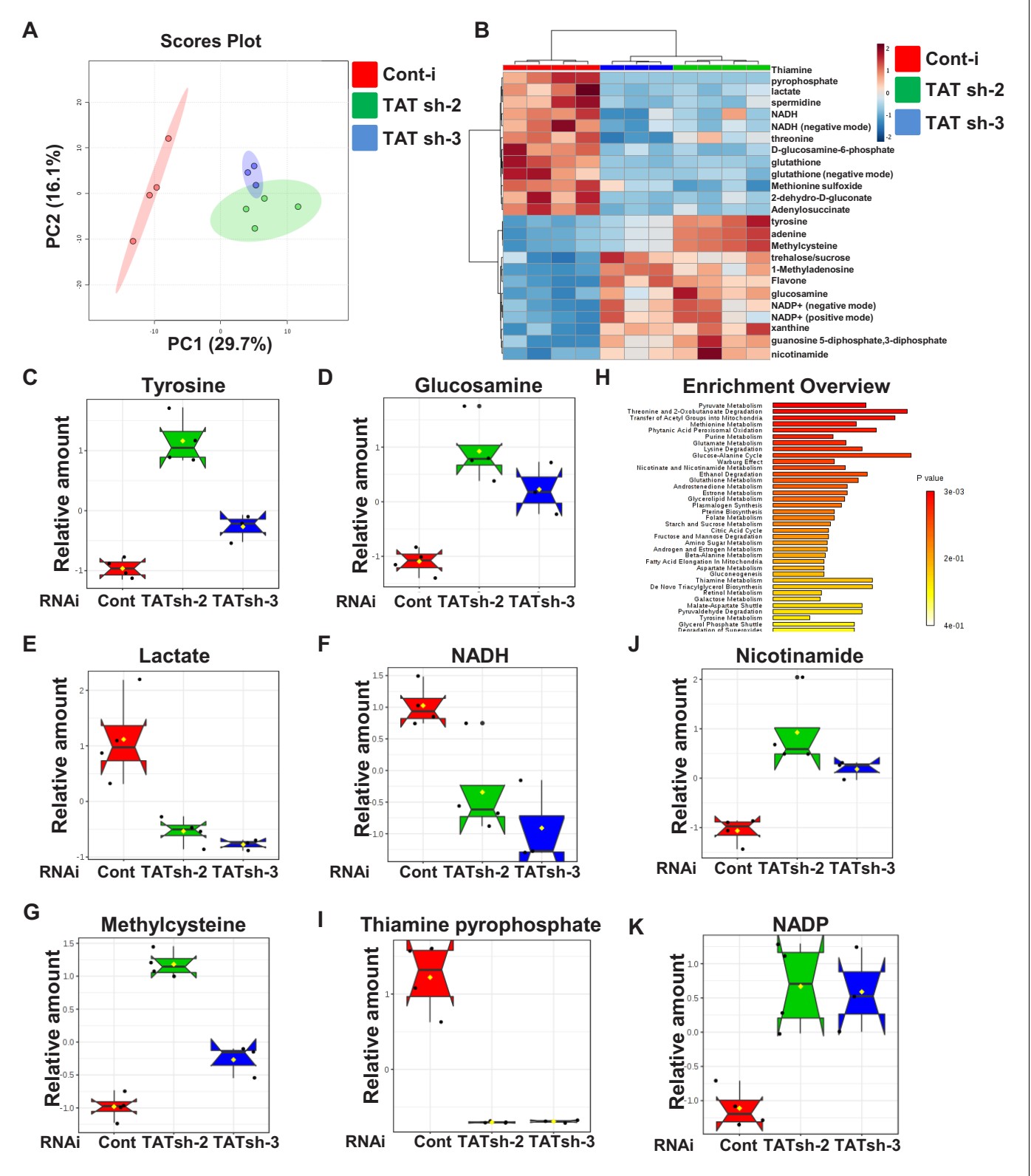

**Figure 4.** Downregulation of CG1461/Tyrosine Aminotransferase leads to reprogramming of metabolism related to mitochondrial function. (**A**) Principal component analysis of *tubulin-Gal4, tubulin-Gal80ts* flies expressing either control RNAi or two different *TAT* RNAi. (**B**) Heat map showing the significantly and commonly changed metabolites in flies expressing two different *TAT* RNAi. Box plots of relative levels of tyrosine (**C**), lactate (**D**), glucosamine (**E**), nicotinamide (**F**), methylcysteine (**G**) in *tubulin-Gal4,tubulin-Gal80ts* flies expressing either control RNAi or two different *TAT* RNAi. (**H**)
*Figure 4 continued on next page*

*Figure 4 continued*

Metabolic Set Enrichment Analysis of the metabolites that changed significantly and commonly in flies expressing two different *TAT* RNAi. Box plots of relative levels of NADH (I), thiamine pyrophosphate (J), NADP (K) in *tubulin-Gal4,tubulin-Gal80ts* flies expressing either control RNAi or two different *TAT* RNAi.

The online version of this article includes the following figure supplement(s) for figure 4:

**Figure supplement 1.** Neuronal-specific downregulation of Tyrosine Aminotransferase/TAT increases levels of tyrosine-derived neurotransmitters.

reprogramming involving metabolites belonging to mitochondrial metabolism and antioxidant defense that have known roles in the regulation of lifespan.

## Whole-body downregulation of *TAT* elevates levels of tyrosine-derived neurotransmitters

Tyrosine is a precursor for biogenic amine neurotransmitters: dopamine, tyramine, and octopamine. Tyrosine can be hydroxylated by tyrosine hydroxylase (TH/*ple*) to produce DOPA, and DOPA can be decarboxylated by Dopa decarboxylase (Ddc) to produce dopamine. Alternatively, tyrosine can be decarboxylated by tyrosine decarboxylase (Tdc1/Tdc2) to produce tyramine. Tyramine can be further converted by tyramine-β-hydroxylase (TβH) to octopamine. Octopamine and tyramine are the invertebrate counterparts of the vertebrate adrenergic transmitters adrenaline and noradrenaline. We hypothesized that redirection of tyrosine from the production of neurotransmitters into the tyrosine degradation pathway could result in either decreased levels of neurotransmitters or aggravation of mitochondrial function via feeding of tyrosine into the TCA cycle. To test whether TAT is involved in the regulation of the levels of tyrosine-derived neurotransmitters, we measured levels of tyrosine, DOPA, Dopamine, Tyramine, and Octopamine in heads of *TAT* wild-type, heterozygous, and mutant middle-age flies when the levels of TAT and other enzymes in the tyrosine degradation pathway are significantly increased. Loss of *TAT* led to the increase of tyrosine (*Figure 5A*) and all tyrosine-derived neurotransmitters (*Figure 5B,C,D,E*), but did not affect the levels of Histamine or GABA (*Figure 4—figure supplement 1A,B*). We further tested whether neuronal-specific downregulation of *TAT* would also increase the levels of tyrosine-derived neurotransmitters. We measured levels of DOPA, Dopamine, Tyramine, and Octopamine in heads of flies that expressed either control RNAi or strong *TAT* RNAi under the control of the neuronal-specific temperature sensitive (*elav-Gal4, tubulin-Gal80ts*) driver for 7 days and that were maintained either on regular food or food containing 5 mM of tyrosine for 2 days before the analysis. Neuronal-specific downregulation of TAT in the combination with supplementation of tyrosine caused upregulation of DOPA and Octopamine (*Figure 4—figure supplement 1C,D*). We have not detected significant increase in the levels of either tyramine or dopamine (*Figure 4—figure supplement 1E,F*), potentially, due to compensatory degradation of tyrosine in non-neuronal tissues or insufficient timing of TAT downregulation.

We then tested whether supplementation of neurotransmitters via feeding can prolong lifespan. It has been previously demonstrated that octopamine-deficient *Tβh*-null flies are sterile because they retain fully developed eggs, a defect that can be rescued by transferring flies onto octopamine- or norepinephrine- supplemented food (*Monastirioti et al., 1996*). Also, intermittent octopamine feeding to adult flies can substitute for exercise in sedentary flies, providing a number of pro-healthspan benefits (*Sujkowski et al., 2017*). Similarly, behavioral defect (sensitization to cocaine) in *iav* mutant flies that have significantly reduced levels of tyramine due to reduced activity of the enzyme tyrosine decarboxylase can be rescued by supplementing the food with tyramine (*McClung and Hirsh, 1999*). In addition, L-DOPA feeding rescues disrupted behaviors in neural dopamine-deficient flies (*Riemensperger et al., 2011*). To test whether age-dependent decrease in the levels of neurotransmitters are related to aging, we fed wild-type *OreR* flies with 5 mM of tyramine, octopamine, and L-DOPA starting at week 2 of age (choosing the concentration that was able to rescue genetic defects associated with loss of these neurotransmitters). Although this supplementation marginally (but statistically significantly) increased *Drosophila* lifespan (*Figure 4—figure supplement 1G,H*), the effect was weaker as compared to the lifespan extension observed following neuronal *TAT* downregulation, suggesting potential involvement of additional mechanisms of lifespan extension by downregulation of *TAT*.

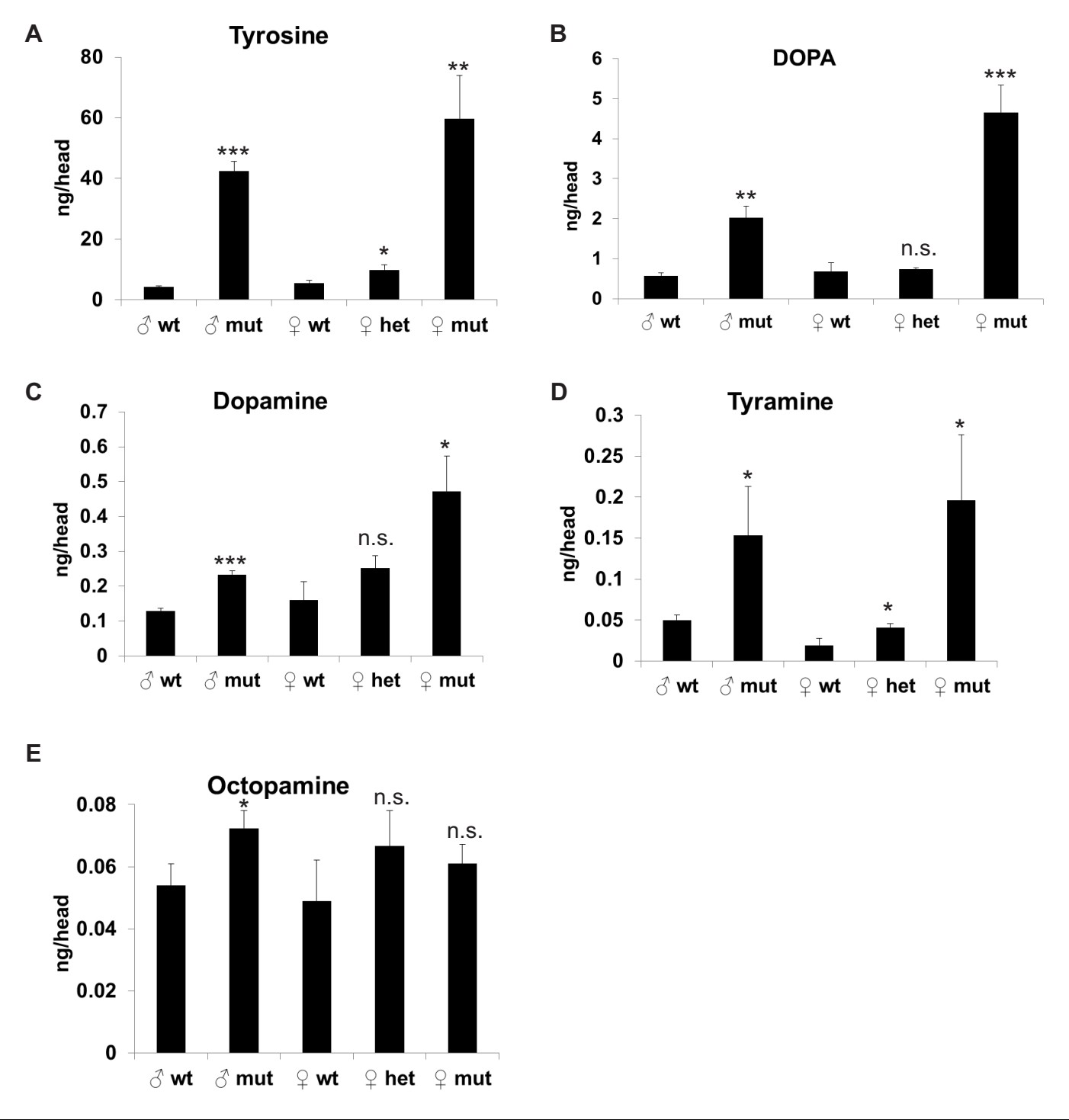

**Figure 5.** Whole-body downregulation of Tyrosine Aminotransferase/TAT elevates levels of tyrosine-derived neurotransmitters in fly heads. Head levels of Tyrosine (**A**), DOPA (**B**), Dopamine (**C**), Tyramine (**D**), and Octopamine (**E**) in *CG1461/TAT* wild-type (wt), heterozygous (het), and mutant (mut) flies. Means ± SD. *p<0.05, **p<0.01, ***p<0.001.

The online version of this article includes the following figure supplement(s) for figure 5:

**Figure supplement 1.** Suppression of complex I of ETC upregulates mRNA levels of enzymes in the tyrosine degradation pathway and decreases lifespan that can be partially rescued by supplementation of tyrosine.

# Inhibition of the electron transport chain upregulates expression of components of the tyrosine degradation pathway that can be rescued by Tigecycline

While most cells use glucose/pyruvate/lactate for ATP synthesis, changes in cellular homeostasis can lead to a switch in fuel utilization that results in the oxidation of fatty acids and amino acids to produce NADH and $FADH_2$ to feed the mitochondrial electron transport chain (ETC) (*Area-Gomez et al., 2019*). Tyrosine can be degraded via the tyrosine degradation pathway and generate two fragments, each of which can enter the TCA cycle. Four of the nine carbon atoms of tyrosine generate free acetoacetate, which is converted into acetoacetyl-CoA, and the second four-carbon fragment is recovered as fumarate. Eight of the nine carbon atoms of these two amino acids thus enter the citric acid cycle and the remaining carbon is lost as $CO_2$. We hypothesized that aging and neurodegeneration can serve as a signal for the switch in tyrosine metabolism from production of neurotransmitters into the tyrosine degradation pathway and further aggravate mitochondrial dysfunction. To test our hypothesis, we expressed RNAi against different components of ETC (CG9762 – Complex I, SDHC – Complex II, CG18809 – Complex IV, and ms [*Hoffman et al., 2014*] 72Dt – Complex V) in young flies under the control of a ubiquitous temperature-sensitive driver (*tubulin-Gal4, tubulin-Gal80ts*) for 10 days. Suppression of all components of ETC resulted in a profound increase of mRNA levels of *TAT* (between 1.5- and two fold induction) with the greatest effect seen with downregulation of *CG9762* (Complex I) (*Figure 6A*). We tested eight additional different subunits of complex I (CG9172, NP15.6, CG8680, CG1970, CG3214, mtacp1). Downregulation of each of them resulted in a similar increase in mRNA levels of *TAT* (up to fivefold induction) (*Figure 6A*). We then tested whether other enzymes in the tyrosine degradation pathway respond to the suppression of complex I of ETC. Similar to *TAT*, downregulation of different subunits of complex I caused a strong increase in mRNA levels of *faa*, *Hgo*, and *CG11796*, enzymes that act downstream of TAT in the tyrosine degradation pathway (*Figure 5—figure supplement 1A*). We also tested whether neuronal-specific suppression of complex I of ETC via downregulation of NP15.6 would phenocopy the effect of the whole body downregulation of NP15.6. We measured levels of *TAT* and other enzymes in the tyrosine degradation pathway in whole flies that expressed either control RNAi or strong *TAT* RNAi under the control of the neuronal-specific temperature-sensitive (*elav-Gal4, tubulin-Gal80ts*) driver for 10 days. Suppression of complex I of ETC only in neuronal cells was not enough to increase the expression of *TAT* or other enzymes in the tyrosine degradation pathway (*Figure 5—figure supplement 1B*). Interestingly, downregulation of components of complex I ETC can either extend or suppress the *Drosophila* lifespan depending on the strength and/or duration of this suppression (*Copeland et al., 2009*; *Foriel et al., 2019*; *Rea et al., 2007*). We further tested whether supplementation of tyrosine to flies with the whole-body downregulation of a component of Complex I ETC - NP15.6 would affect their lifespan. Expression of *NP15.6* RNAi under the control of the ubiquitous temperature sensitive (*tubulin-Gal4, tubulin-Gal80ts*) driver caused significant reduction of lifespan with a stronger effect in female flies ($-13\%$ in male flies, $p<0.0001$, log-rank test; $-33\%$ in female flies, $p<0.0001$, log-rank test) (*Figure 5—figure supplement 1C,D*). Supplementation of different concentrations of tyrosine partially rescued lifespan in both male ($+8.7\%$, $+11\%$, $+11\%$ for 1X, 2.5X, and 5X concentrations of tyrosine, $p<0.0001$, log-rank test) (*Figure 5—figure supplement 1C*) and female flies ($+9.5\%$, $+10\%$, $+10.5\%$ for 1X, 2.5X, 5X concentrations of tyrosine, $p<0.0001$, $p=0.0016$, $p=0.0002$, log-rank test) (*Figure 5—figure supplement 1D*). In summary, inhibition of mETC Complex I function can decrease lifespan, which can be partially rescued by tyrosine supplementation.

There is extensive evidence for the involvement of mitochondrial dysfunction in the pathogenesis of neurodegenerative diseases such as Alzheimer's disease (AD) and Parkinson's disease (PD) (*Area-Gomez et al., 2019*). Thus, we further tested whether pathological conditions associated with mitochondrial dysfunction would also cause the upregulation of TAT. To investigate whether expression of α-synuclein previously shown to promote mislocalization of the mitochondrial fission protein Drp1 leading to mitochondrial dysfunction and neuronal death also induced expression of *TAT*, we used a *Drosophila* α-synucleinopathy model (*Ordonez et al., 2018*). As expected and similar to ETC complex I inhibition, expression of wild-type human α-synuclein using the Syb-QF2 panneuronal driver resulted in the increase of *TAT* mRNA levels (*Figure 6B*). We further tested whether downregulation of *TAT* would rescue the neuronal loss associated with α-synuclein expression. We

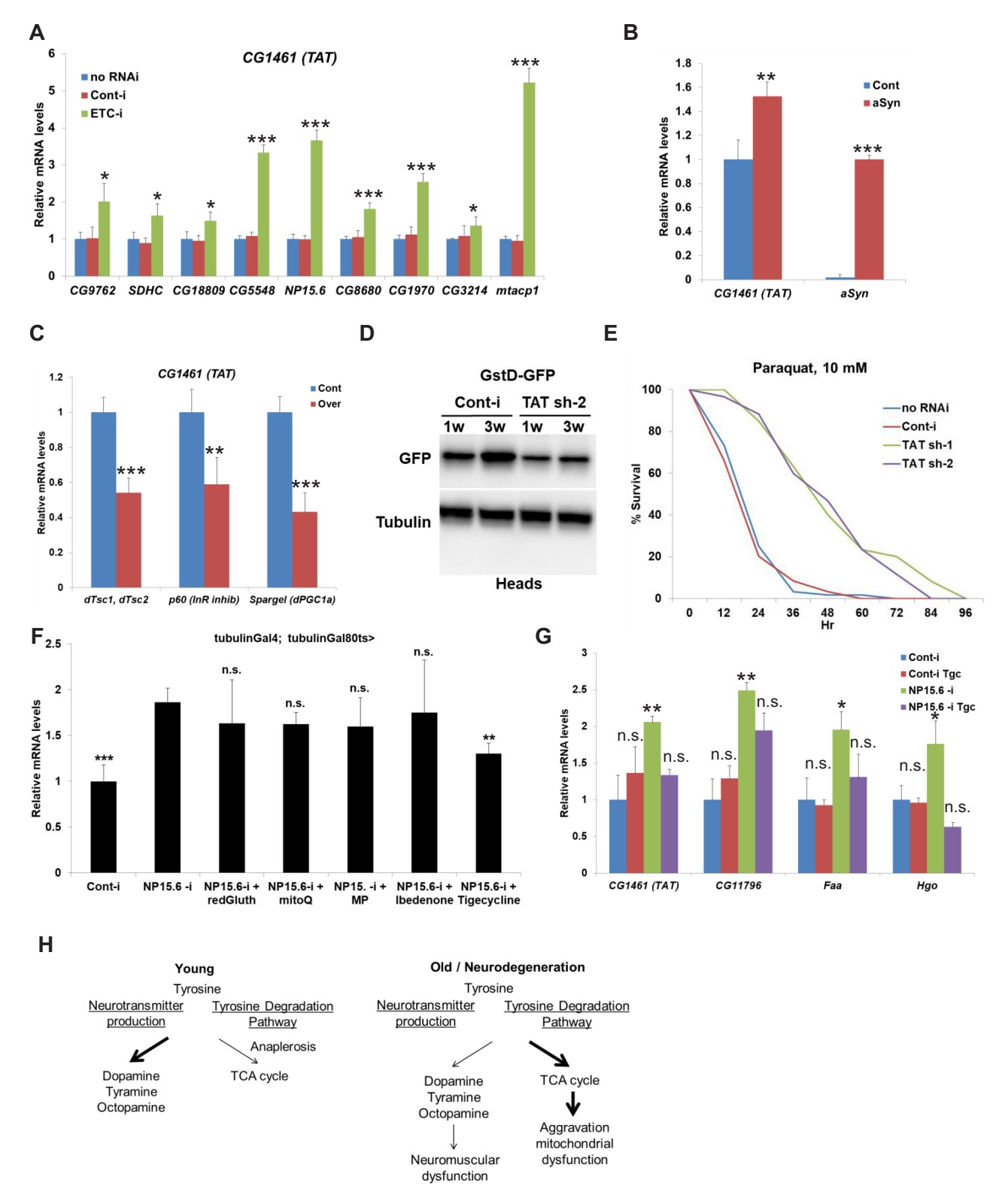

**Figure 6.** Mitochondrial dysfunction/neurodegeneration upregulates the level of CG1461/tyrosine aminotransferase. (**A**) Relative mRNA levels of *CG1461/TAT* in *Gal80ts; tubulin-Gal4* flies expressing either no RNAi, control RNAi or RNAi against different subunits of mitochondrial ETC – *CG9762, SDHC, CG18809, CG5548, NP15.6, CG8680, CG1970, CG3214, mtacp1* for 10 days. Means ± SD. (**B**) Relative mRNA levels of *CG1461/TAT* and α-synuclein in heads of flies with or without expression of wild-type human α-synuclein. Means ± SD. (**C**) Relative mRNA levels of *CG1461/TAT* in *tubulin-*

*Figure 6 continued*

*Gal80ts, tubulin-Gal4* flies overexpressing control or p60 (inhibitory subunit of InR), dTsc1/dTsc2 (TSC complex, dTOR inhibitor) or PGC1a/Spargel. Means ± SD. (**D**) Immunoblot analysis of GFP and tubulin in heads of 1-week and 3-week-old flies expressing either control or *TAT* RNAi under pan-neuronal driver (ElavGal4) in the presence of GstD-GFP. (**E**) Ubiquitous adult-onset downregulation of TAT prolongs lifespan under oxidative stress (10 mM Paraquat). (**F**) Relative mRNA levels of *CG1461/TAT* in *tubulin-Gal80ts, tubulin-Gal4* flies expressing either control or NP15.6 RNAi and fed with 10 mM reduced Glutathione, 100 µM mitoQ, 10 mM methyl pyruvate, 100 µM Ibedenone, or 100 µM Tigecycline. Means ± SD (**G**) Relative mRNA levels of *CG1461/TAT*, *CG11796*, *faa*, and *Hgo* in *tubulin-Gal80ts, tubulin-Gal4* flies expressing either control or NP15.6 RNAi and fed with either control or 100 µM Tigecycline. (**H**) Working model. *p<0.05, **p<0.01, ***p<0.001.

The online version of this article includes the following figure supplement(s) for figure 6:

**Figure supplement 1.** The effect of downregulation of tyrosine aminotransferase/TAT on GFP-tagged reporters relevant to aging.

downregulated *TAT* using three different RNAi lines in the presence of α-synuclein expression. As expected, α-synuclein expression caused strong neurodegeneration and dramatic neuronal loss; however, downregulation of TAT with three different RNAi lines did not suppress this phenotype (*Figure 5—figure supplement 1E,F*). PD is characterized by extensive reprogramming of metabolism and targeting of multiple metabolic pathways may be required to prevent neurodegeneration (*Shao and Le, 2019*).

To test whether stimulation of mitophagy/mitochondrial biogenesis decreases the level of TAT in young flies, we tested how overexpression of *Spargel* (*Drosophila* orthologue of *PGC1α*), *p60* (inhibitory subunit of InR), or *dTsc1/dTsc2* (TSC complex, dTOR inhibitor) affects the level of TAT. Both overexpression of *Spargel* (*Rera et al., 2011*) and suppression of InR (*Clancy et al., 2001*) or TOR (*Kapahi et al., 2004*) signaling are critical regulators of mitophagy/mitochondrial biogenesis and extend *Drosophila* lifespan (*Figure 6—figure supplement 1A*). Overexpression of *Spargel*, *p60,* and *dTsc1/dTsc2* significantly suppressed expression of *TAT* (*Figure 6C*). To test how mechanistically downregulation of *TAT* extends lifespan, we crossed flies expressing either control RNAi or *TAT* RNAi with a pan-neuronal driver (ElavGal4) to a number of GFP reporter lines relevant to different age-dependent processes—GstD-GFP (oxidative stress) (*Sykiotis and Bohmann, 2008*), GFP-CL1 (proteosomal activity) (*Pandey et al., 2007*), hsp22-GFP (mitochondrial stress) (*Yang and Tower, 2009*), STAT92E-GFP (Stat pathway) (*Bach et al., 2007*), and Drs-GFP (antimicrobial response) (*Ferrandon et al., 1998*)—and tested their activity in fly heads. We found that the amount of GFP under the control of the *GstD* promoter increased with age in fly brains, and this increase was suppressed by downregulation of *TAT* (*Figure 6D*),while we have not detected differences between control and *TAT* RNAi with other reporters, at least at the tested conditions (*Figure 6—figure supplement 1B–E*). This suggests that downregulation of *TAT* may regulate lifespan at least partially via counteracting the accumulation of oxidative stress in neuronal tissue. We further found that downregulation of *TAT* strongly increased resistance to 10 mM paraquat (redox cycler which causes oxidative damage) (*Figure 6E*). Downregulation of ETC complex I can have multiple effects on mitochondria leading to increased mitoROS production, mtUPR, decreased NAD+/NADH ratio, decreased ATP production and other effects (*Rhooms et al., 2020*). To dissect a potential mechanism of how downregulation of ETC Complex I and potentially aging cause upregulation of *TAT*, we fed flies with an available toolbox of chemical inhibitors: mitoQ dye (quencher of mitochondrial ROS), methyl pyruvate (membrane permeable pyruvate), idebenone (a bypass of mitochondrial complex I), Tigecycline (antibiotic that can inhibit mitochondrial translation), and reduced glutathione (antioxidant) (*Wisnovsky et al., 2016*). Only treatment with Tigecycline suppressed upregulation of *TAT* induced by inhibition of ETC Complex I (*Figure 6F*), although we cannot rule out that other chemicals did not rescue *TAT* upregulation because of technical limitations associated with feeding flies with chemicals (poor distribution in fly body, wrong concentration, instability in fly food, etc.). The effect observed for Tigecycline suggests that downregulation of ETC Complex I and potentially aging causes an integrated stress response that can be rescued by inhibition of mitochondrial protein translation. Moreover, treatment with Tigecycline also suppressed other enzymes in the tyrosine degradation pathway (*CG11796, Hgo, Faa*) that were upregulated by the inhibition of ETC Complex I but did not affect their levels in control flies (*Figure 6G*). Interestingly, although inhibition of mitochondrial translation would be expected to cause mitochondrial stress, at this concentration/timing, we have not observed upregulation of markers of mitochondrial stress (*ClpX, hsp10, hsp60,* or *hsp22*) in both control flies and in flies with inhibition of ETC Complex I (*Figure 6—figure*

*supplement 1F*). Altogether, our results suggest that, similar to aging, mitochondrial dysfunction serves as a signal to elevate expression of *TAT* and other enzymes in the tyrosine degradation pathway and that this effect can be rescued by the FDA-approved drug Tigecycline.

## Discussion

By comparing metabolic changes in control and long-lived flies during aging, we identified the amino acid tyrosine and the tyrosine degradation pathway as potential targets for lifespan extension. Supplementing flies with tyrosine or whole-body and tissue-specific downregulation of tyrosine aminotransferase/TAT, the first and rate-limiting enzyme in the tyrosine degradation pathway, significantly extended lifespan. Moreover, upregulation of TAT might serve as a switch between neurotransmitter production and anaplerosis under mitochondrial dysfunction and aging. In addition, we found that the FDA-approved drug Tigecycline can suppress the upregulation of enzymes in the tyrosine degradation pathway under the conditions of mitochondrial dysfunction.

### Age-dependent metabolic reprogramming

We searched for changes in the fly metabolome caused by aging in control and long-lived flies and hypothesized that preventing some of these changes would increase lifespan and prevent age-dependent health deterioration. Previously, we found striking differences in multiple methionine metabolism intermediates between control and long-lived flies, including S-adenosylhomocysteine and homocysteine, and showed that modulating levels of these metabolites extended lifespan (*Parkhitko et al., 2016*). Here, we extended our analysis and identified 49 metabolites that changed significantly with age between control and long-lived flies. Some of these metabolites represent metabolic pathways that were previously implicated in lifespan extension and the role of others is unknown. Systematic interrogation of the enzymes regulating these metabolites should lead to identification of previously unknown regulators of lifespan in *Drosophila*.

### Amino acid metabolism and lifespan extension

Amino acids play a key role in lifespan extension by calorie/dietary restriction. In *Drosophila*, restricting dietary protein extends lifespan, whereas carbohydrate and lipid restriction have little effect on survival (*Mair et al., 2005*). In addition, manipulation of metabolism of specific amino acids can extend lifespan in flies and other organisms. For example, methionine restriction (*Lee et al., 2014*) or activation of the methionine flux (*Parkhitko et al., 2016*; *Parkhitko et al., 2019*) prolongs health- and lifespan in flies and other species, whereas the mechanisms of lifespan extension do not overlap. This effect could be due to the processing of harmful metabolites, as activation of the methionine flux would promote processing of SAH, which accumulates with age and inhibits methyltransferases (*Parkhitko et al., 2019*). Impairing threonine catabolism by downregulation of glycine-C-acetyltransferase promotes the lifespan of *C. elegans* via stimulation of methylglyoxal formation (*Ravichandran et al., 2018*). Methylglyoxal is a reactive dicarbonyl inducing oxidative stress and while its high concentration is harmful, low concentration stimulates stress-responsive pathways and hormetic activity on lifespan in *C. elegans* (*Ravichandran et al., 2018*). Increased homocysteine processing via the transsulfuration pathway (*Kabil et al., 2011*) and glycine supplementation extend *Drosophila* and *C. elegans* lifespan, respectively, partially via stimulating the flux through methionine metabolism. Here, we identified a new branch of amino acid metabolism, namely the tyrosine degradation pathway that can regulate lifespan. In agreement with our data showing that levels of TAT are decreased in flies with overexpression of *dTsc1/Tsc2*, Yuan et al. found that the level of *hpd-1*, the worm orthologue of 4-hydroxyphenylpyruvate dioxygenase/CG11796, is decreased in long-lived *eat-2* mutant worms (genetic model of calorie restriction in worms). Moreover, downregulation of *hpd-1* increased worm lifespan, which is consistent with increased *Drosophila* lifespan observed with downregulation of *CG11796* (*Yuan et al., 2012*). In addition, the *C. elegans* ortholog of *TAT*, *tatn-1*, influences insulin signaling, development, and lifespan via modulation of aak-2/AMPK signaling (*Ferguson et al., 2013*). Also, in worms, AMPK/calcineurin-mediated longevity was regulated cell-nonautonomously via regulation of octopamine (*Burkewitz et al., 2015*). Altogether, we propose two potential mechanisms of lifespan extension: preservation of the production of neurotransmitters and prevention of increased tyrosine feeding into the TCA cycle (*Figure 6H*).

## Amino acid catabolism and mitochondria

Aging is characterized by progressive accumulation of dysfunctional mitochondria and a reduced capacity to produce ATP. Mitochondrial ETC generates a proton gradient derived from NADH and $FADH_2$ that are produced in the TCA cycle, and ATP synthase uses this gradient for the production of ATP from ADP. Changes in cellular homeostasis can result in the oxidation of amino acids to produce NADH and $FADH_2$ to feed the mitochondrial electron transport chain (ETC) (*Area-Gomez et al., 2019*). Due to the fact that: (a) metabolism of different amino acids has recently been shown to play an important role in the regulation of aging; (b) mitochondrial dysfunction is evident in aging and neurodegeneration; (c) tyrosine can play anaplerotic role via the tyrosine degradation pathway and production of acetoacetyl CoA and fumarate; and (d) enzyme levels in the tyrosine degradation pathway increase with age, we hypothesize that, at least in flies, aging drives upregulation of tyrosine catabolism and redirects tyrosine from neurotransmitter production into the tyrosine degradation pathway to compensate for age-driven mitochondrial dysfunction (*Figure 6H*). Moreover, we demonstrate that these changes can be phenocopied by the inhibition of mETC with the strongest effect seen with inhibition of mETC complex I. Although it is unknown how the activity of the mETC complex I is changed with age in neuronal cells in flies, complex I activity declines with age in both rodents and human in different organs including brain (*Gómez and Hagen, 2012*). Moreover, overexpression of *NDI1*, an alternative NADH dehydrogenase that can bypass complex I, prolonged *Drosophila* lifespan when it was either expressed in whole flies or only in neuronal cells (*Sanz et al., 2010*; *Bahadorani et al., 2010*), raising the question of how downregulation of mETC complex I extends lifespan while it can also phenocopy aging and reduce lifespan. It has been shown in worms that lifespan extension caused by inhibition of ETC is limited by the narrow window of timing of treatment and strength of knockdown (*Rea et al., 2007*). Similarly in flies, lifespan extension by inhibition of mETC was achieved with the da-GeneSwitch driver, which is weaker and mosaic compared with non-inducible Gal4 drivers (*Copeland et al., 2009*), whereas usage of stronger and ubiquitous drivers caused severe lifespan shortening (*Foriel et al., 2019*). It is still an open question why inhibition of mETC is beneficial in one context and detrimental in another.

## Tyrosine aminotransferase and tyrosinemia

Tyrosinemia type II, also known as Richner-Hanhart Syndrome (OMIM # 276600), is a rare autosomal recessive disorder, caused by mutations in the gene encoding tyrosine aminotransferase and is manifested by eye, skin, and central nervous system alterations (*Scott, 2006*). Why would downregulation of TAT extend lifespan in flies but cause disease in humans? The degree of downregulation, duration, and tissue specificity could be a key for the beneficial effects of TAT downregulation. Indeed, we observed stronger lifespan extension with nervous-specific *TAT* downregulation than with ubiquitous ActinGeneSwitch driver. In addition, *TAT*-mutant flies had shortened lifespan. Moreover, the ubiquitous ActinGeneSwitch driver has a mosaic pattern of expression that would probably still allow degradation of some tyrosine and create a balance between neurotransmitter production and tyrosine degradation. This notion is consistent with the fact that patients heterozygous for *TAT* do not exhibit any clinical manifestations. In agreement with the beneficial role of downregulation of TAT on lifespan, our metabolomics analysis revealed upregulation of several metabolites previously connected to lifespan extension. These metabolites include GlcN-6-phosphate (*Weimer et al., 2014*), nicotinamide (*Balan et al., 2008*; *Liu et al., 2018*), and methylcysteine (*Wassef et al., 2007*). Targeting metabolic pathways attributed to these metabolites extend health- and lifespan across different species (*Parkhitko et al., 2020*). It will be of interest to elucidate the mechanisms leading to the elevation of these metabolites. NTBC/nitisinone/Orfadin (2-(2-nitro-4-trifluoromethylbenzoyl)cyclohexane-1,3-dione) is an FDA-approved drug that inhibits 4-hydroxyphenylpyruvate dioxygenase in the tyrosine degradation pathway and is currently approved for the treatment of hereditary tyrosinemia type 1 (*Lock et al., 2014*). Given that our current study shows that inhibition of the tyrosine degradation pathway and specifically of CG11796, the fly ortholog of 4-hydroxyphenylpyruvate dioxygenase, extends lifespan, it would be of interest to test the potential of NTBC/nitisinone/Orfadin to promote health- and lifespan in humans.

### Tyrosine supplementation in human clinical trials

Tyrosine is a precursor for biogenic amine neurotransmitters: dopamine, tyramine, and octopamine. Tyramine and octopamine in flies are analogous to epinephrine (adrenaline) and norepinephrine (noradrenaline) in humans. In mice, tyrosine supplementation reaches maximum concentration in the brain 1 hr after oral ingestion (*Topall and Laborit, 1989*) and enhances catecholamine synthesis in particular noradrenergic neurons (*Gibson and Wurtman, 1978*; *Milner and Wurtman, 1986*). In rats, tyrosine supplementation increases dopamine concentration in the extracellular fluid of corpus striatum and nucleus accumbens (*During et al., 1988*; *Woods and Meyer, 1991*). Due to the beneficial role of catecholamines in coping with stress, a number of clinical trials in healthy subjects have been designed to investigate whether tyrosine reduces cognitive and physiological stress in humans exposed to a combination of mental and physical stressors. In humans, tyrosine has been shown to improve cold-induced decrements in working memory (*Shurtleff et al., 1994*), improve subject performance on stress-sensitive attention tasks (*Deijen and Orlebeke, 1994*), improve attentional focus in the presence of a distractor (*Deijen and Orlebeke, 1994*), and attenuate performance decrements and adverse mood states associated with acute exposure to cold and hypoxia (*Deijen and Orlebeke, 1994*; *Banderet and Lieberman, 1989*; *Dollins et al., 1995*; *Shukitt-Hale et al., 1996*). Most of these clinical trials in healthy subjects have been performed on young volunteers. Interestingly, the first study investigating the effects of tyrosine on cognitive effects in older adults has demonstrated unfavorable effects of higher doses tyrosine on working memory performance (*van de Rest et al., 2017*). These findings are consistent with our hypothesis that tyrosine metabolism is reprogrammed in older adults and that at this stage tyrosine supplementation enhances the tyrosine degradation pathway instead of enhancing the production of neurotransmitters. Finally, it would be interesting to test whether Tigecycline, an FDA-approved drug that suppresses the activity of the tyrosine degradation pathway in flies in response to mitochondrial dysfunction could restore normal levels of tyrosine-derived neurotransmitters in elderly people.

## Materials and methods

### Key resources table

| Reagent type (species) or resource | Designation | Source or reference | Identifiers | Additional information |
|---|---|---|---|---|
| Antibody | Anti-α-Tubulin | Sigma | T5168 | |
| Antibody | Anti-GFP | Invitrogen | A-6455 | |
| Strain, strain background (*Escherichia coli*) | One Shot TOP10 Chemically Competent *E. coli* | Thermo Fisher Scientific | C404003 | |
| Commercial assay or kit | iScript Reverse Transcription Supermix | Bio-Rad | 1708896 | |
| Commercial assay or kit | iQ SYBR Green Supermix | Bio-Rad | 1708880 | |
| Commercial assay or kit | BenchMark Prestained Protein Ladder | Invitrogen | 10748–010 | |
| Chemical compound, drug | TRIzol reagent | Invitrogen | 15596–018 | |
| Chemical compound, drug | Mifepristone | Cayman Chemical Company | 10006317 | |
| Chemical compound, drug | Methyl viologen dichloride hydrate (Paraquat) | Sigma-Aldrich | 856177 | |
| Chemical compound, drug | ProSieve EX transfer buffer | Lonza | 00200309 | |
| Chemical compound, drug | ProSieve EX running buffer | Lonza | 200307 | |
| Chemical compound, drug | L-Glutathione reduced | Sigma-Aldrich | G6013-5G | |

*Continued on next page*

*Continued*

| Reagent type (species) or resource | Designation | Source or reference | Identifiers | Additional information |
|---|---|---|---|---|
| Chemical compound, drug | Laemmli Sample Buffer | Bio-Rad | 1610737 | |
| Chemical compound, drug | RIPA | Cell Signaling | 9806 | |
| Chemical compound, drug | Protease Inhibitor Cocktail Tablets | Roche | 4693159001 | |
| Chemical compound, drug | Idebenone | Cayman Chemical Company | 15475 | |
| Chemical compound, drug | Mitoquinol | Cayman Chemical Company | 89950 | |
| Chemical compound, drug | Nuclease-Free Water (not DEPC-Treated) | Ambion, Inc | AM9930 | |
| Chemical compound, drug | 3,4-Dihydroxy-L-phenylalanine | Millipore Sigma | D9628 | |
| Chemical compound, drug | Octopamine hydrochloride | Millipore Sigma | O0250 | |
| Chemical compound, drug | RQ1 RNase-Free DNase | Promega | M6101 | |
| Chemical compound, drug | Tyramine | Sigma-Aldrich | T90344 | |
| Commercial assay or kit | 4–20% Mini-PROTEAN TGX Precast Protein Gels | Bio-Rad | 4561095 | |
| Commercial assay or kit | pENTR/D-TOPO Cloning Kit | Life Technologies | K2400-20 | |
| Commercial assay or kit | Gateway LR Clonase II Enzyme mix | Invitrogen | 11791–020 | |
| Genetic reagent (*D. melanogaster*) | *white* RNAi (HMS00017) | Bloomington *Drosophila* Stock Center | # 33623 | |
| Genetic reagent (*D. melanogaster*) | *GFP* RNAi (HMS00314) | Perrimon's lab | | |
| Genetic reagent (*D. melanogaster*) | *CG1461* RNAi (HMC03212) | Bloomington *Drosophila* Stock Center | # 51470 | |
| Genetic reagent (*D. melanogaster*) | *CG1461* RNAi (HMS05690) | Bloomington *Drosophila* Stock Center | # 67830 | |
| Genetic reagent (*D. melanogaster*) | *CG1461* RNAi (HMS05877) | Bloomington *Drosophila* Stock Center | # 76065 | |
| Genetic reagent (*D. melanogaster*) | *Hgo* RNAi (HMC03775) | Bloomington *Drosophila* Stock Center | # 55629 | |
| Genetic reagent (*D. melanogaster*) | *CG11796* RNAi (HMC03663) | Bloomington *Drosophila* Stock Center | # 52923 | |
| Genetic reagent (*D. melanogaster*) | *CG9762* RNAi (HMC06415) | Bloomington *Drosophila* Stock Center | # 67311 | |
| Genetic reagent (*D. melanogaster*) | *Sdhc* RNAi (HMC03497) | Bloomington *Drosophila* Stock Center | # 53281 | |
| Genetic reagent (*D. melanogaster*) | *CG18809* RNAi (HMS04326) | Bloomington *Drosophila* Stock Center | # 56907 | |

*Continued on next page*

*Continued*

| Reagent type (species) or resource | Designation | Source or reference | Identifiers | Additional information |
|---|---|---|---|---|
| Genetic reagent (*D. melanogaster*) | *CG5548* RNAi (HM05255) | Bloomington *Drosophila* Stock Center | # 30511 | |
| Genetic reagent (*D. melanogaster*) | *NP15.6* RNAi (HMS01560) | Bloomington *Drosophila* Stock Center | # 36672 | |
| Genetic reagent (*D. melanogaster*) | *CG8680* RNAi (HMC03434) | Bloomington *Drosophila* Stock Center | # 51860 | |
| Genetic reagent (*D. melanogaster*) | *CG1970* RNAi (HMC04814) | Bloomington *Drosophila* Stock Center | # 57499 | |
| Genetic reagent (*D. melanogaster*) | *CG3214* RNAi (HMS01584) | Bloomington *Drosophila* Stock Center | # 36695 | |
| Genetic reagent (*D. melanogaster*) | *mtacp1* RNAi (HM05206) | Bloomington *Drosophila* Stock Center | # 29528 | |
| Genetic reagent (*D. melanogaster*) | B3 | Gift from Dr. Trudy Mackay | | |
| Genetic reagent (*D. melanogaster*) | O1 | Gift from Dr. Trudy Mackay | | |
| Genetic reagent (*D. melanogaster*) | O3 | Gift from Dr. Trudy Mackay | | |
| Genetic reagent (*D. melanogaster*) | OregonR | Perrimon's lab | | |
| Genetic reagent (*D. melanogaster*) | tubulinGal4; tubulinGal80ts | Perrimon's lab | | |
| Genetic reagent (*D. melanogaster*) | CG1461-mutant | This paper | | |
| Genetic reagent (*D. melanogaster*) | Actin-GeneSwitch-Gal4 | Gift from Dr. John Tower | | |
| Genetic reagent (*D. melanogaster*) | Whole body fat body – GeneSwitch – Gal4 | Gift from Dr. John Tower | | |
| Genetic reagent (*D. melanogaster*) | TIGS-2 | Gift from Dr. John Tower | | |
| Genetic reagent (*D. melanogaster*) | 3X Elav-GeneSwitch-Gal4 | Gift from Dr. Scott Pletcher | | |
| Genetic reagent (*D. melanogaster*) | ElavGal4; tubulinGal80ts | Perrimon's lab | | |
| Genetic reagent (*D. melanogaster*) | UAS-dTsc1,Tsc2 | *Tapon et al., 2002* | | |
| Genetic reagent (*D. melanogaster*) | UAS-p60 | Bloomington *Drosophila* Stock Center | # 25899 | |
| Genetic reagent (*D. melanogaster*) | UAS-Spargel | Gift from Dr. David Walker | | |
| Genetic reagent (*D. melanogaster*) | GstD-GFP | Gift from Dr. Dirk Bohmann | | |
| Genetic reagent (*D. melanogaster*) | CG1461-GFP-tagged | VDRC stock center | # 318640 | |
| Genetic reagent (*D. melanogaster*) | GFP-CL1 | Gift from Dr. Udai Pandey | | |

*Continued on next page*

*Continued*

| Reagent type (species) or resource | Designation | Source or reference | Identifiers | Additional information |
|---|---|---|---|---|
| Genetic reagent (*D. melanogaster*) | hsp22-GFP | Gift from Dr. John Tower | | |
| Genetic reagent (*D. melanogaster*) | STAT92-GFP | Bloomington *Drosophila* Stock Center | # 26198 | |
| Genetic reagent (*D. melanogaster*) | Drs-GFP | Bloomington *Drosophila* Stock Center | # 55707 | |
| Genetic reagent (*D. melanogaster*) | Xbp1-GFP | Bloomington *Drosophila* Stock Center | # 60730 | |
| Genetic reagent (*D. melanogaster*) | Syb-QF2; QUAS- α-synuclein | Gift from Dr. Mel Feany | | |

## Lifespan analysis

For survival analysis, flies were collected within 24 hr from eclosion, sorted by sex under light $CO_2$ anesthesia, and reared at standard density (20–25 flies per vial) on cornmeal/soy flour/yeast fly food at 25°C and 60% humidity with 12 hr on/off light cycle. Flies were transferred to fresh vials every 2 days and dead flies counted. RU486 dissolved in ethanol was administered in the media at the final concentration of 150 ug/mL. The following RNAi lines were used: *white* RNAi (HMS00017); *GFP* RNAi (HMS00314); *CG1461/TAT/tyrosine aminotransferase* RNAi [TAT-1 HMC03212 (weak), TAT-2 HMS05690 (strong), TAT-3 HMS05877 (strong)], *Hgo* RNAi (HMC03775), *CG11796* RNAi (HMC03663), *CG9762* RNAi (HMC06415), *Sdhc* RNAi (HMC03497), *CG18809* RNAi (HMS04326), *CG5548* RNAi (HM05255), *NP15.6* RNAi (HMS01560), *CG8680* RNAi (HMC03434), *CG1970* RNAi (HMC04814), *CG3214* RNAi (HMS01584), *mtacp1* RNAi (HM05206). All lifespan counts are listed in the *Supplementary file 1*.

## qRT-PCR

Total RNA was extracted with the TRIzol reagent (Life Technologies), followed by DNase digestion using RQ1 RNase-Free DNase (Promega). Total RNA was reverse transcribed with the iScript cDNA synthesis kit (Bio-Rad). qRT-PCR was performed with the iQ SYBR Green Supermix (Bio-Rad) and a CFX96 Real- Time PCR Detection System (Bio-Rad). *RpL32* and *alpha-Tubulin 84B* were used as a normalization reference. Relative quantitation of mRNA levels was determined with the comparative $C_T$ method.

| Gene | FlyPrimerBank ID | Forward primer | Reverse primer |
|---|---|---|---|
| *TAT* | PP16388 | CGCTGTCCATTGGTGATCCC | TGGCATACCCATTGTACTTGC |
| *TAT* | PP28777 | GGCTCCAAGCTATCCCTTAACA | CACCAATGGACAGCGGTATCA |
| *Faa* | PP37488 | GGGATGTGGTAAGAAGCCAGA | CTGTGGCACAATGGAAACCG |
| *Hgo* | PP36016 | CTTCCATTCCAGCCCTTCAAG | TTTCCATCCTTTGGAGGCAAG |
| *CG11796* | PP24189 | GGATTGCCCTCCACCAAGC | CAGGATTCTCTCGTACCAGGA |
| *GstZ2* | PP18005 | CCGCGAGGTGAATCCAATG | CTGGGGACGTGTTTCCTCC |
| *Hn* | PP19433 | TGTTTTCGCCCAAGGATTCGT | CACCAGGTTTATGTCATGCTTCT |
| *aSynuclein* | | AAGAGGGTGTTCTCTATGTAGGC | GCTCCTCCAACATTTGTCACTT |
| *Rp49* | | ATCGGTTACGGATCGAACAA | GACAATCTCCTTGCGCTTCT |

## Antibodies and immunoblot analyses

A rabbit anti-GFP antibody was obtained from Abcam, and the anti-tubulin antibody from Sigma-Aldrich. For immunoblot analyses, 10 flies or 20 heads were grinded in bead beader in RIPA (Cell

Signaling) lysis buffer with phosphatase and protease inhibitors (Roche). Whole-cell lysates were resolved by electrophoresis, and proteins were transferred onto PVDF membranes (Immobilon P; Millipore), blocked in Tris-buffered saline Tween-20 buffer (Cell Signaling Technology) containing 2.5% dry milk, and probed with the indicated antibodies diluted in this buffer.

## Statistical analysis
Statistical analyses were performed in either JMP (SAS, Cary, NC, USA) or Excel.

## Metabolite profiling
Ten to 20 flies per sample (four biological replicates) were collected and intracellular metabolites extracted using 80% (v/v) aqueous methanol. A 5500 QTRAP hybrid triple quadrupole mass spectrometer (AB/SCIEX) coupled to a Prominence UFLC HPLC system (Shimadzu) was used for steady-state analyses of the samples. Selected reaction monitoring (SRM) of 287 polar metabolites using positive/negative switching with HILIC chromatography was performed. Peak areas from the total ion current for each metabolite SRM transition were integrated using MultiQuant v2.1 software (AB/SCIEX). The resulting raw data from the MultiQuant software were analyzed using MetaboAnalyst (http://www.metaboanalyst.ca/MetaboAnalyst/).

## Quantification of neurotransmitters from fly heads
Five fly heads per sample (three biological replicates) were homogenized in 190 µL of acidified acetone, 10 µL of 10 mM ascorbic acid and 1 µL of a 5 µg/mL mixture of the corresponding deuterated internal standards (CDN Isotopes, Quebec, Canada). The supernatants were collected, dried, and derivatized using in-house synthesized 6-aminoquinolyl-N-hydroxysuccinimidyl carbamate (AQC). Sample cleanup was done using solid phase extraction (SPE) cartridges (Phenomenex, Inc Hyderabad, India). The eluted samples were completely dried before reconstitution in 2% acetonitrile with 0.5% formic acid and injection into the UHPLC ESI-MS. The liquid chromatography (LC) gradient and electro-spray ionization (ESI) conditions were as described in *Ramesh and Brockmann, 2019*. The calibration curves were made in neat solvents over a 32-fold concentration range. The highest concentration on column for the different compounds: Tyrosine 64 ng, DOPA 0.16 ng, Dopamine 1.6 ng, Tyramine 0.32 ng, Octopamine 1.6 ng, Histamine 3.2 ng, GABA 160 ng. Quantitation was done using the Xcalibur software (version 2.2 SP1.48).

## Neuron counts
Neurons were counted according to the protocol developed by *Ordonez et al., 2018*; *Olsen and Feany, 2019*. Flies were fixed in formalin, embedded in paraffin, and 2 µm serial frontal sections were prepared through the entire fly brain. Slides were processed through xylene, ethanols, and into water and stained with hematoxylin. Images of the medulla brain region were taken at 40x magnification using the SPOT software. The neurons were counted and normalized to a pixel aspect ratio of 2.6 pixels/µm in the FIJI (ImageJ) software. For each fly genotype, six brains were imaged and analyzed.

## Acknowledgements
We thank Trudy Mackay for providing the B3, O1 and O3 flies, John Tower for the GeneSwitch fly stocks and hsp22-GFP line, Scott Pletcher for providing the TIGS-2 and 3XElav Gene-Switch driver lines, David Walker for providing UAS-dPGC-1/Spargel line, Mel Feany for providing Syb-QF2; QUAS- α-synuclein line, Udai Pandey for GFP-CL1 line, Dirk Bohmann for GstD-GFP line. We thank Jay Hirsh for comments on measuring neurotransmitters and Kit Tuen for excellent technical assistance. We thank Stephanie Mohr for critical reading of the manuscript. We thank the TRiP at Harvard Medical School supported by NIH/NIGMS R01-GM084947 for providing transgenic RNAi lines and NCBS institutional fund to AB (12P4167) to support the quantification of neurotransmitters. This work was supported by NIA K99/R00 AG057792 (AP), 5P01CA120964 (NP) and AFAR (NP). NP is an investigator of the Howard Hughes Medical Institute.

# Additional information

## Funding

| Funder | Grant reference number | Author |
| --- | --- | --- |
| National Institute on Aging | K99/R00 AG057792 | Andrey A Parkhitko |
| National Institutes of Health | 5P01CA120964 | Norbert Perrimon |
| National Centre for Biological Sciences | 12P4167 | Axel Brockmann |
| American Foundation for Aging Research | | Norbert Perrimon |

The funders had no role in study design, data collection and interpretation, or the decision to submit the work for publication.

## Author contributions

Andrey A Parkhitko, Conceptualization, Data curation, Formal analysis, Funding acquisition, Investigation, Writing - original draft, Writing - review and editing; Divya Ramesh, Formal analysis, Investigation, Methodology; Lin Wang, Investigation, Methodology; Dmitry Leshchiner, Elizabeth Filine, Richard Binari, Investigation; Abby L Olsen, Axel Brockmann, Supervision, Methodology; John M Asara, Resources, Investigation, Methodology; Valentin Cracan, Conceptualization; Joshua D Rabinowitz, Conceptualization, Methodology; Norbert Perrimon, Conceptualization, Resources, Supervision, Funding acquisition, Project administration, Writing - review and editing

## Author ORCIDs

Andrey A Parkhitko (iD) https://orcid.org/0000-0001-9852-8329
Divya Ramesh (iD) http://orcid.org/0000-0003-1387-7832
Lin Wang (iD) https://orcid.org/0000-0002-9370-6891
Valentin Cracan (iD) https://orcid.org/0000-0002-3280-0593
Axel Brockmann (iD) https://orcid.org/0000-0003-0201-9656
Norbert Perrimon (iD) https://orcid.org/0000-0001-7542-472X

## Decision letter and Author response

Decision letter https://doi.org/10.7554/eLife.58053.sa1
Author response https://doi.org/10.7554/eLife.58053.sa2

# Additional files

## Supplementary files

- Supplementary file 1. Lifespan counts are recorded for all experiments.

- Transparent reporting form

## Data availability

All data generated or analyzed during this study are included in the manuscript and supporting files. Additional metabolomics data are available in our previous publication (https://doi.org/10.1101/gad.282277.116).

The following datasets were generated:

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
