## [Decision Letter]

**Acceptance summary:**

This study demonstrates that aging and mitochondrial dysfunction upregulates the tyrosine degradation pathway and its downregulation extends lifespan. Parkhitko et al. demonstrate how reversing the age related increase in the tyrosine degradation enzyme, TAT, extends lifespan and enhances oxidative stress resistance.

**Decision letter after peer review:**

Thank you for submitting your article "Aging and mitochondrial dysfunction upregulate the tyrosine degradation pathway and its downregulation extends lifespan" for consideration by *eLife*. Your article has been reviewed by three peer reviewers, including Pankaj Kapahi as the Reviewing Editor and Reviewer #1, and the evaluation has been overseen by Jessica Tyler as the Senior Editor. The following individual involved in review of your submission has agreed to reveal their identity: David Walker (Reviewer #2).

The reviewers have discussed the reviews with one another and the Reviewing Editor has drafted this decision to help you prepare a revised submission. As the editors have judged that your manuscript is of interest, but as described below that additional experiments are required before it is published, we would like to draw your attention to changes in our revision policy that we have made in response to COVID-19 (https://elifesciences.org/articles/57162). First, because many researchers have temporarily lost access to the labs, we will give authors as much time as they need to submit revised manuscripts. We are also offering, if you choose, to post the manuscript to bioRxiv (if it is not already there) along with this decision letter and a formal designation that the manuscript is "in revision at *eLife*".

Summary:

In "Aging and mitochondrial dysfunction upregulate the tyrosine degradation pathway and its downregulation extends lifespan," Parkhitko et al. demonstrate that reversing the age related increase in the tyrosine degradation enzyme, TAT, extends lifespan. They suggest that inhibition of ETC function, which changes with age, alters tyrosine metabolism.

The scientific rationale of the study is novel and the study is technically well performed to support the hypothesis. However, there are a number of issues and questions could be addressed to make the study stronger.

Reviewer #1:

In "Aging and mitochondrial dysfunction upregulate the tyrosine degradation pathway and its downregulation extends lifespan," Parkhitko et al. demonstrate how reversing the age related increase in the tyrosine degradation enzyme, TAT, extends lifespan. The scientific rationale of the study is novel and the study is technically well performed to support the hypothesis. However, there are a few issues and questions could be addressed to make the study stronger.

Figure 3 demonstrates that TAT knockdown in the neurons extends lifespan, and that whole body knockdown might not be as strong due to detrimental effects in other tissues. However, the later figures revert back to exploring the effects of TAT knockdown in whole body. As such, could the authors describe how Figures 4, 5 and 6E would be impacted with Elav-driven TAT RNAi in addition to whole body.

In Figure 6 the authors put forward the hypothesis that aging and neurodegeneration can serve as a signal for the switch in tyrosine metabolism from production of neurotransmitters into the tyrosine degradation pathway to compensate for age-dependent functional decline, which would further aggravate mitochondrial dysfunction. However, several questions remain to be answered to validate this hypothesis.

What happens to the ETC-I during normal aging in the long-lived flies shown in Figure 1?

Are the ETC-I -RNAi flies short lived? There are fly studies with opposite effects too. Importantly for this study, can this shortening be rescued by increasing tyr levels and knockdown of TAT.

Does loss of ETC-I cell-autonomously drive TAT expression?

The final figure claims a link of TAT activity to neurodegeneration and PD, which is an interesting aspect of the study that would broaden its significance. Could the authors support their findings by examining neurodegeneration in WT and PD strains to support these arguments.

I have concerns about the tigecycline results as it is also known to cause mitochondrial dysfunction and oxidative stress. Furthermore, several assays described above need to be done with tigecycline thus I suggest to remove it from this study unless it can be supported further.

Reviewer #2:

This is a very interesting new paper from Parkhitko et al.

The major strength of the work is the use of an unbiased “omics” approach to identify a novel pathway (Tyrsosine degradation) relevant to aging and lifespan determination in *Drosophila*. The findings outlined in the paper will be of interest to a general readership and will likely inspire additional work in this new area.

In terms of potential experimental limitations to the work, it seems that the authors have not provided an "RU486 control" for any of the experimental work. This is potentially important as the Tower and Ja labs have reported that depending upon the food used and RU conc. used there can be unpredictable effects of RU feeding on fly longevity. e.g., J Gerontol A Biol Sci Med Sci. 2017 Feb;72(2):173-180.

Hence, it is strongly advised to include the following control for longevity studies:

control strain/GS Driver line +/- RU (same conc. as used with transgene of interest).

I note, however, that the authors tested several RNAi lines and not all produced positive effects. Therefore, the observed effects are unlikely due to RU feeding alone. In addition, it is always sensible to assay food intake in the context of longevity-promoting interventions. This also seems to be missing from the study, which could complicate interpretation of the findings.

Suggested edits and perceived impact of the work:

In an ideal world (without a pandemic), at least some of the suggestions below could be addressed with experiments. But, given current events, these issues could be further discussed in the manuscript:

It is interesting and potentially informative that Tyrosine levels go up in long-lived flies. This may reflect a midlife increase, with “midlife” being a different chronological age in different genetic backgrounds. Presumably, even in long-lived strains tyrosine levels decrease in late life?

Can the genetic interventions reported prolong lifespan when the interventions are started in mid- or even late-life? If so, this would significantly increase the perceived impact of the work.

The relationship with ETC knockdown and tyrosine degradation is interesting. The authors report that ETC knockdown (mito dysfunction) phenocopies aging. However, as the authors note in the Introduction: moderate ETC knockdown can prolong lifespan in flies (Copeland et al.,; Owusu-Ansah et al.) and worms. Different outcomes (lifespan extensions vs lifespan shortening) are likely a question of timing and level of knock-down and this may be worth discussing further. On face value, however, this appears paradoxical. Hence, more discussion may benefit the readers.

The work showing that tigecycline prevents activation of the tyrosine degradation pathway, upon mito dysfunction, is an additional strength of the paper. However, it raises an obvious follow up question: can tigecycline be used to treat aging WT flies in a similar manner?

If so, this would significantly increase the impact of the work. If the drug could be used to slow aging/prolong lifespan upon treatment of aged WT flies. Can the drug rescue the decrease in tyrosine levels in WT aged flies?

Reviewer #3:

Parkhitko studies the impact of TAT on lifespan, metabolites and reporter phenotypes. The work contrasts metabolomic profiles of the Rose "B" and "O" lines and shows different age patterns of tyrosine among the lines. This leads to a putative tyrosine aminotransferase (TAT/CG1461). Lifespan is extended when TAT is reduced in adults. The authors record many correlated phenotypes of TAT RNAi flies, ranging from metabolomics to neurotransmitters. They find inhibition of mitochondrial ETC increases tyrosine degradation elements. Some data are indeed interesting but conclusions on mechanisms of aging are overly extrapolated and untested. The work has substantial problems in terms of inference, design and scientific rigor.

Summary of substantive concerns:

1) Comparing the "B" and "O" lines is difficult to interpret – issues explained years back by many authors but here unaware. Fundamentally, the B lines are not “unselected controls”. Before Rose, they were strongly selected for precocious development and reproductive schedule, and they also accumulated substantial mutational load for genes expressed soon after eclosion. The O lines swept out the load and selected back toward a normal reproductive schedule. Traits among the lines can show age-dependent differences but have no bearing on aging.

2) It is not sufficient to measure only two ages. That tyrosine increases between 1 and 4w in O lines (or 1 and 5w in OreR) is incomplete. Perhaps tyrosine increases until 2w then declines thereafter, yet is higher at 4w than at 1w. Aging is a progressive function.

3) O lines ramp up fecundity only after a few weeks. The B lines start from eclosion. It is quite possible that tyrosine metabolism reflects reproductive activity, not progress of somatic aging.

4) We need to see the absolute amount of tyrosine at 1 and 4w for each B and O line. Using “relative amount” obscures.

5) In many places the paper generates correlative data but packages interpretations as causality. For instance, no data establish "downregulation of TAT causes metabolic reprogramming by affecting mitochondrial/antioxidant pro-longevity metabolic factors". The data show that downregulation of TAT induces metabolic changes, some of which are associated with “mitochondrial/antioxidant factors”. The work does not establish TAT acts through mitochondria to cause the observed metabolic changes, or that this affects aging. Overall, the work requires epistasis analyses.

6) It is interesting to show that inhibition of ETC upregulates TAT mRNA (and other components), which can be rescued by a drug to block ETC damage. No data establish a role for ETC via tyrosine metabolism to extend survival. Likewise, to “test how mechanistically downregulation of TAT extends lifespan” only measures stress GFP reporters, not lifespan.

7) The neuropeptide work is incomplete. Here, is the drug assimilated? If not, a negative effect is meaningless. Yet, L-DOPA does improve survival. The “marginal and significant” but “much weaker” (subjective call) survival increase is actually within the range of the TAT RNAi data, if we can interpret the survival plot with its poor dynamics. And the drug study was not replicated. These data do not rule out a role of elevated L-DOPA.

8) There are many inconsistent uses of which RNAi strain, driver, and gender combinations to make any one point. Why, or what data is not being shown?

---

## [Author Response]

The scientific rationale of the study is novel and the study is technically well performed to support the hypothesis. However, there are a number of issues and questions could be addressed to make the study stronger.Reviewer #1:In "Aging and mitochondrial dysfunction upregulate the tyrosine degradation pathway and its downregulation extends lifespan," Parkhitko et al. demonstrate how reversing the age related increase in the tyrosine degradation enzyme, TAT, extends lifespan. The scientific rationale of the study is novel and the study is technically well performed to support the hypothesis. However, there are a few issues and questions could be addressed to make the study stronger.Figure 3 demonstrates that TAT knockdown in the neurons extends lifespan, and that whole body knockdown might not be as strong due to detrimental effects in other tissues. However, the later figures revert back to exploring the effects of TAT knockdown in whole body. As such, could the authors describe how Figures 4, 5 and 6E would be impacted with Elav-driven TAT RNAi in addition to whole body.

Based on our model, we propose that downregulation of TAT prevents redirection of tyrosine from the degradation pathway to the production of neuromediators, and inhibition of this redirection in neuronal cells is enough to extend lifespan. However, these neuromediators could have pro-health and pro-longevity effects throughout the whole body. In Figure 4, we demonstrate that whole-body driven TAT knockdown causes wholebody metabolic changes associated with increased lifespan. In Figure 5, we demonstrate that TAT mutant flies have increased levels of neuromediators, and in Figure 6 we demonstrate that whole-body knockdown of ETC components upregulates levels of enzymes in the tyrosine degradation pathway. To address the reviewer’s question, we measured the levels of neurotransmitters in heads of flies with neuronal-specific *Elav*-Gal4-driven TAT downregulation. Neuronal-specific downregulation of TAT in combination with supplementation of tyrosine causes upregulation of dopa and octopamine. We have not detected significantly increased levels of either tyramine or dopamine potentially due to degradation of tyrosine in non-neuronal tissues. Similarly, neuronal-specific downregulation of TAT caused the upregulation of tyrosine and metabolic reprogramming similar to whole-body downregulation of TAT. We detected several metabolites (tyrosine, glucosamine, methylcysteine, and NADH) that were commonly and significantly altered as a result of whole-body-driven and neuronal-specific downregulation of TAT. In contrast, neuronal-specific inhibition of mETC Complex I has not resulted in the whole-body upregulation of TAT or other enzymes in the tyrosine degradation pathway. Based on our model, mitochondrial dysfunction leads to the redirection of tyrosine metabolism into the tyrosine degradation pathway that causes alteration in the production of neuromediators, and neuromediators cause nonautonomous effects on other tissues in the organism. It is also possible, that mitochondrial dysfunction can reprogram tyrosine metabolism in other organs non-autonomously, an issue that would be interesting to investigate in follow up studies.

In Figure 6 the authors put forward the hypothesis that aging and neurodegeneration can serve as a signal for the switch in tyrosine metabolism from production of neurotransmitters into the tyrosine degradation pathway to compensate for age-dependent functional decline, which would further aggravate mitochondrial dysfunction. However, several questions remain to be answered to validate this hypothesis.What happens to the ETC-I during normal aging in the long-lived flies shown in Figure 1?

In our original manuscript we demonstrated that inhibition of other ETC complexes also caused the upregulation of the enzymes in the tyrosine degradation pathway but the effect of ETC-I inhibition was the strongest. Although it is unknown how the activity of mETC complex I is changed with age in neuronal cells in flies, complex I activity declines with age in both rodents and human in different organs including the brain that we now mention. We also added additional references demonstrating that bypass of ETC-1 extends lifespan and similar to our studies it had a neuronal-specific effect. Regarding a comparison of ETC function in B and O flies, we cannot perform this experiment as we do not have the original flies any longer. These flies that we originally received from Dr. Trudy Mackay were maintained under special selection regimen (kept in population cages and only eggs from 1 month old flies would be transferred to fresh cages to prevent selection for fast reproduction) and are no longer available as the MacKay lab did not maintain them under this special selection regimen.

Are the ETC-I -RNAi flies short lived? There are fly studies with opposite effects too. Importantly for this study, can this shortening be rescued by increasing tyr levels and knockdown of TAT.

This is a very context-dependent question. We describe in the Introduction that downregulation of components of ETC extends *Drosophila* lifespan (Copeland et al.). Moreover, in the Copeland et al. paper downregulation of CG9762 extends lifespan and we demonstrate that downregulation of CG9762 upregulates mRNA levels of TAT. However, there are also *Drosophila* models of Complex I deficiency that are characterized by decreased lifespan. In addition, bypass of Complex I via overexpression of Ndi1 has been shown to extend *Drosophila* lifespan. Also, in worms, suppression of ETC extends lifespan only under very specific circumstances including timing of treatment and strength of knockdown. In the Copeland et al. paper, the authors used the *da-GS-Gal4* driver, while in our study we used *tubulin-Gal4, tubulin-Gal80ts* driver. The two studies are quite different, especially as GeneSwitch drivers are in general weaker and mosaic. We discussed these differences in the Discussion. We used *tubulin-Gal4, tubulin-Gal80ts* driver to suppress Complex I via expression of NP15.6 RNAi and maintained flies at different concentrations of tyrosine.

Downregulation of NP15.6 strongly decreased lifespan in females but had a weaker effect in males. Feeding flies with different concentrations of tyrosine partially rescued their decreased lifespan. Interestingly, these gender-specific differences correlate with sex-specific differences in expression of the components of the tyrosine degradation pathway.

Does loss of ETC-I cell-autonomously drive TAT expression?

In worms, octopamine mediates cell-nonautonomous effects of AMPK/calcineurin-mediated longevity (Burkewitz et al.) and ETC knockdown in neurons affects mitochondrial homeostasis in distal tissues (Durieux et al.). It is a great area for the future investigations whether ETC downregulation and tyrosine metabolism alterations interact via cell-autonomous or cell-nonautonomous mechanisms. We have not detected wholebody upregulation of TAT or other enzymes in the tyrosine degradation pathway as a result of neuronal-specific inhibition of mETC Complex I. However, it does not exclude the possibility of non-autonomous interactions between inhibition of mETC Complex I and the regulation of the tyrosine degradation pathway.

The final figure claims a link of TAT activity to neurodegeneration and PD, which is an interesting aspect of the study that would broaden its significance. Could the authors support their findings by examining neurodegeneration in WT and PD strains to support these arguments.

To address the reviewer’s comment we performed histology on control and PD flies with control or three different *TAT* RNAi lines. As expected, expression of α-Synuclein caused strong neurodegeneration and loss of neurons. However, downregulation of *TAT* did not prevent the neuronal loss. α-Synuclein expression potentially causes extensive reprogramming of metabolism and it may require targeting of multiple metabolic pathways to prevent neurodegeneration.

I have concerns about the tigecycline results as it is also known to cause mitochondrial dysfunction and oxidative stress. Furthermore, several assays described above need to be done with tigecycline thus I suggest to remove it from this study unless it can be supported further.

We expect the tigecycline effects (similar to inhibition of Complex I of ETC) to be very context/time dependent. For example, tigecycline has been shown to have very different effects between normal and cancer cells. To address whether tigecycline causes upregulation of the markers of mitochondrial dysfunction, we tested several markers of mitochondrial stress by qRT-PCR, *CLPX, HSP10, HSP60, and HSP22*. The concentration and duration of tigecycline that prevents upregulation of *TAT* does not cause upregulation of markers of mitochondrial dysfunction.

Reviewer #2:This is a very interesting new paper from Parkhitko et al.The major strength of the work is the use of an unbiased “omics” approach to identify a novel pathway (Tyrsosine degradation) relevant to aging and lifespan determination in *Drosophila*. The findings outlined in the paper will be of interest to a general readership and will likely inspire additional work in this new area.In terms of potential experimental limitations to the work, it seems that the authors have not provided an "RU486 control" for any of the experimental work. This is potentially important as the Tower and Ja labs have reported that depending upon the food used and RU conc. used there can be unpredictable effects of RU feeding on fly longevity. e.g., J Gerontol A Biol Sci Med Sci. 2017 Feb;72(2):173-180.Hence, it is strongly advised to include the following control for longevity studies:control strain/GS Driver line +/- RU (same conc. as used with transgene of interest).I note, however, that the authors tested several RNAi lines and not all produced positive effects. Therefore, the observed effects are unlikely due to RU feeding alone. In addition, it is always sensible to assay food intake in the context of longevity-promoting interventions. This also seems to be missing from the study, which could complicate interpretation of the findings.

We tested all GeneSwitch lines crossed to control strains in our previous paper (Parkhitko et al., 2016). In this paper, we also tested and published that doubling of the concentration of RU486 does not affect the lifespan in our hands.

Suggested edits and perceived impact of the work:In an ideal world (without a pandemic), at least some of the suggestions below could be addressed with experiments. But, given current events, these issues could be further discussed in the manuscript:It is interesting and potentially informative that Tyrosine levels go up in long-lived flies. This may reflect a midlife increase, with “midlife” being a different chronological age in different genetic backgrounds. Presumably, even in long-lived strains tyrosine levels decrease in late life?

We agree with this statement. We included only 1w and 4w flies because we wanted to avoid the actively dying state. The goal was only to get potential metabolic pathways as candidates for lifespan extension.

Can the genetic interventions reported prolong lifespan when the interventions are started in mid- or even late-life? If so, this would significantly increase the perceived impact of the work.

We agree that it would have a strong translational impact. We added this to the text. However, one caveat is that GeneSwitch Gal4 lines may not be as active later in life as in the beginning (based on unpublished discussions with other people in aging field), which could make the interpretation of results difficult.

The relationship with ETC knockdown and tyrosine degradation is interesting. The authors report that ETC knockdown (mito dysfunction) phenocopies aging. However, as the authors note in the Introduction: moderate ETC knockdown can prolong lifespan in flies (Copeland et al.,; Owusu-Ansah et al.) and worms. Different outcomes (lifespan extensions vs lifespan shortening) are likely a question of timing and level of knock-down and this may be worth discussing further. On face value, however, this appears paradoxical. Hence, more discussion may benefit the readers.

We expanded this part in the Discussion. See also comments to reviewer #1.

The work showing that tigecycline prevents activation of the tyrosine degradation pathway, upon mito dysfunction, is an additional strength of the paper. However, it raises an obvious follow up question: can tigecycline be used to treat aging WT flies in a similar manner?If so, this would significantly increase the impact of the work. If the drug could be used to slow aging/prolong lifespan upon treatment of aged WT flies. Can the drug rescue the decrease in tyrosine levels in WT aged flies?

It is an interesting question. Tigecycline is a mitochondrial translation inhibitor. We do not expect that long-term treatment with tigecycline would prolong lifespan because of prolonged mitochondrial stress – an issue also raised by reviewer #1. In the timeframe that we used to treat flies control flies and flies with inhibition of ETC CI we did not observe activation of markers of mitochondrial stress. However, it would be a great separate project to study how inhibition of mitochondrial translation interplays with aging and production of neuromediators.

Reviewer #3:Parkhitko studies the impact of TAT on lifespan, metabolites and reporter phenotypes. The work contrasts metabolomic profiles of the Rose "B" and "O" lines and shows different age patterns of tyrosine among the lines. This leads to a putative tyrosine aminotransferase (TAT/CG1461). Lifespan is extended when TAT is reduced in adults. The authors record many correlated phenotypes of TAT RNAi flies, ranging from metabolomics to neurotransmitters. They find inhibition of mitochondrial ETC increases tyrosine degradation elements. Some data are indeed interesting but conclusions on mechanisms of aging are overly extrapolated and untested. The work has substantial problems in terms of inference, design and scientific rigor.Summary of substantive concerns:1) Comparing the "B" and "O" lines is difficult to interpret – issues explained years back by many authors but here unaware. Fundamentally, the B lines are not “unselected controls”. Before Rose, they were strongly selected for precocious development and reproductive schedule, and they also accumulated substantial mutational load for genes expressed soon after eclosion. The O lines swept out the load and selected back toward a normal reproductive schedule. Traits among the lines can show age-dependent differences but have no bearing on aging.

We used these lines only to get a list of candidate metabolic pathways that can potentially be relevant to aging. Next, we used this information to validate changes in levels of enzymes belonging to the tyrosine degradation pathway in wild-type flies, flies with inhibition of ETC CI, and in flies with downregulation of enzymes belonging to the tyrosine degradation pathway.

2) It is not sufficient to measure only two ages. That tyrosine increases between 1 and 4w in O lines (or 1 and 5w in OreR) is incomplete. Perhaps tyrosine increases until 2w then declines thereafter, yet is higher at 4w than at 1w. Aging is a progressive function.

We decided to choose these two time points because in both B and O lines 100% of the flies are still alive. We did not aim to use metabolomics to explain differences between these strains; our goal was only to get potential candidates for testing in wild-type flies. In addition, we could only run a limited number of samples at the given time and our metabolomics platform does not allow combining results from independent runs. We agree that it could be an interesting and independent project to characterize age-dependent metabolic changes in these lines.

3) O lines ramp up fecundity only after a few weeks. The B lines start from eclosion. It is quite possible that tyrosine metabolism reflects reproductive activity, not progress of somatic aging.

As mentioned above, we expanded the analysis of the tyrosine degradation pathway to fly lines that are independent of B/O lines.

4) We need to see the absolute amount of tyrosine at 1 and 4w for each B and O line. Using “relative amount” obscures.

This point was also raised by another reviewer. Unfortunately, the commercially available kits did not work in our hands on fly samples and we don’t have these lines anymore.

5) In many places the paper generates correlative data but packages interpretations as causality. For instance, no data establish "downregulation of TAT causes metabolic reprogramming by affecting mitochondrial/antioxidant pro-longevity metabolic factors". The data show that downregulation of TAT induces metabolic changes, some of which are associated with “mitochondrial/antioxidant factors”. The work does not establish TAT acts through mitochondria to cause the observed metabolic changes, or that this affects aging. Overall, the work requires epistasis analyses.

We agree with this but did not want to over-interpret the data. We only stated that downregulation of *TAT* causes metabolic alterations that are known to be associated with increased lifespan. We agree that epistatic analysis would be very helpful but because GeneSwitch lines are in general weak Gal4 drivers, we did not want to use them for the longevity experiments in flies. In addition, we and others found that the addition of extra UAS constructs dilutes Gal4 and can weaken phenotypes making results difficult to interpret, in addition of mixing genetic backgrounds.

6) It is interesting to show that inhibition of ETC upregulates TAT mRNA (and other components), which can be rescued by a drug to block ETC damage. No data establish a role for ETC via tyrosine metabolism to extend survival. Likewise, to “test how mechanistically downregulation of TAT extends lifespan” only measures stress GFP reporters, not lifespan.

To partially address this comment, we tested how downregulation of one of the components of ETC Complex I that causes strong upregulation of TAT affects the lifespan and whether these changes can be rescued by supplementing tyrosine.

7) The neuropeptide work is incomplete. Here, is the drug assimilated? If not, a negative effect is meaningless. Yet, L-DOPA does improve survival. The “marginal and significant” but “much weaker” (subjective call) survival increase is actually within the range of the TAT RNAi data, if we can interpret the survival plot with its poor dynamics. And the drug study was not replicated. These data do not rule out a role of elevated L-DOPA.

We fed flies with concentrations of drugs that have been previously reported to rescue genetic defects associated with production of these neurotransmitters and we used concentrations reported in these publications. We agree that insufficient rescue can still be related to poor drug uptake, drug instability, etc.,. We expanded the Discussion on these potential caveats. The drug study was done on a sufficient number of flies to detect even small changes and lifespans are performed in multiple vials independently and the data on survival are combined.

8) There are many inconsistent uses of which RNAi strain, driver, and gender combinations to make any one point. Why, or what data is not being shown?

In general, we use GeneSwitch lines to perform lifespans and *TubulinGal4/ElavGal4,Gal80ts* lines to perform biochemical analysis because GeneSwitch lines are mosaic and weaker. For the metabolomics analysis, we could run only a limited number of samples and cannot combine results from separate runs. Due to this limitation, we performed metabolomics only for female flies. Because the mitochondrial dysfunction and CI inhibition results originated from the metabolomics studies, we performed CI inhibition experiments on female flies. We have not shown data on multiple sensors to prevent overcrowding the figures but since another reviewer also requested this we have now added the missing data.